# Artificial intelligence based writer identification generates new evidence for the unknown scribes of the Dead Sea Scrolls exemplified by the Great Isaiah Scroll (1QIsa[a])

**Mladen Popović**[1]*, **Maruf A. Dhali**[2], **Lambert Schomaker**[2]

**1** Qumran Institute, Faculty of Theology and Religious Studies, University of Groningen, Groningen, The Netherlands, **2** Department of Artificial Intelligence, Faculty of Science and Engineering, University of Groningen, Groningen, The Netherlands

☯ These authors contributed equally to this work.
* m.popovic@rug.nl

**Data Availability Statement:** All relevant data are within the paper and its Supporting information files.

## Abstract

The Dead Sea Scrolls are tangible evidence of the Bible's ancient scribal culture. This study takes an innovative approach to palaeography—the study of ancient handwriting—as a new entry point to access this scribal culture. One of the problems of palaeography is to determine writer identity or difference when the writing style is near uniform. This is exemplified by the Great Isaiah Scroll (1QIsa[a]). To this end, we use pattern recognition and artificial intelligence techniques to innovate the palaeography of the scrolls and to pioneer the micro-level of individual scribes to open access to the Bible's ancient scribal culture. We report new evidence for a breaking point in the series of columns in this scroll. Without prior assumption of writer identity, based on point clouds of the reduced-dimensionality feature-space, we found that columns from the first and second halves of the manuscript ended up in two distinct zones of such scatter plots, notably for a range of digital palaeography tools, each addressing very different featural aspects of the script samples. In a secondary, independent, analysis, now assuming writer difference and using yet another independent feature method and several different types of statistical testing, a switching point was found in the column series. A clear phase transition is apparent in columns 27–29. We also demonstrated a difference in distance variances such that the variance is higher in the second part of the manuscript. Given the statistically significant differences between the two halves, a tertiary, post-hoc analysis was performed using visual inspection of character heatmaps and of the most discriminative Fraglet sets in the script. Demonstrating that two main scribes, each showing different writing patterns, were responsible for the Great Isaiah Scroll, this study sheds new light on the Bible's ancient scribal culture by providing new, tangible evidence that ancient biblical texts were not copied by a single scribe only but that multiple scribes, while carefully mirroring another scribe's writing style, could closely collaborate on one particular manuscript.

**Funding:** 'Mladen Popović Project: The Hands that Wrote the Bible Grant Number: ERC Starting Grant 640497 European Research Council https://erc.europa.eu/ The funders had no role in study design, data collection and analysis, decision to publish, or preparation of the manuscript.'

**Competing interests:** The authors have declared that no competing interests exist.

# 1 Introduction

Ever since their modern discovery, the Dead Sea Scrolls are famous for containing the oldest manuscripts of the Hebrew Bible (Old Testament) and many hitherto unknown ancient Jewish texts. The manuscripts date from the 4th century BCE to the 2nd century CE. They come from the caves near Qumran and other Judaean Desert sites west near the Dead Sea, except for Wadi Daliyeh which is north of Jericho [1]. Among other things, the scrolls provide a unique vantage point for studying the latest literary evolutionary phases of what was to become the Hebrew Bible. As archaeological artifacts, they offer tangible evidence for the Bible's ancient scribal culture 'in action'.

A crucial but hardly used entry point into the Bible's ancient scribal culture is that of individual scribes [2]. There is, however, a twofold problem with putting this entry point of individual scribes into effective use. Except for a handful of named scribes in a few documentary texts [3, 4], the scribes behind the scrolls are anonymous. This is especially true for the scrolls from Qumran, which, with almost a thousand manuscripts of mostly literary texts, represents the largest find site.

The next best thing to scribes identified by name is scribes identified by their handwriting. Although some of the suggestions for a change of scribal hands in a single manuscript or scribes who copied more than one manuscript [5–8] have met with broader assent, most have not been assessed at all. And estimations of the total number of scribes [9–11], an argument in the discussion about the origin of the scrolls from Qumran [4, 12–14], have been, at best, educated guesses.

One of the main problems regarding traditional palaeography of the Dead Sea Scrolls, and also for writer identification in general [15, 16], is the ability to distinguish between variability within the writing of one writer and similarity in style—but with subtle variations—between different writers. On the one hand, scribes may show a range in a variety of forms of individual letters in one or more manuscripts. On the other hand, different scribes might write in almost the same way, making it a challenge to identify the individual scribe beyond general stylistic similarities.

The question is whether perceived differences in handwriting are significant and the result of there being two different writers or insignificant because they are the result of normal variations within the handwriting of the same writer. The problem with knowing which differences are likely to be idiographic, and thus significant, is that, in the end, this also involves using implicit criteria that are experience-based [15, 17]. In this regard, although they work according to differing methodologies [15, 17–19], there is no difference between professional forensic document examiners and palaeographers. The problem is also how one can convince others [20, 21], whether through pictorial form, verbal descriptions, palaeographic charts or a combination thereof.

The Great Isaiah Scroll from Qumran Cave 1 (1QIsaᵃ) exemplifies the lack of a robust method in Dead Sea Scrolls palaeography for how to determine and verify writer identity or difference, especially when the handwriting is near uniform. The question for 1QIsaᵃ is whether subtle differences in writing should be regarded as normal variations in the handwriting of one scribe or as similar scripts of two different scribes and, if the latter, whether the writing of the two scribes coincides with the two halves of the manuscript.

The scroll measures 7.34 m in length, averages 26 cm in height, and contains 54 columns of Hebrew text. There is a codicological caesura between columns 27 and 28 in the form of a three-line lacuna at the bottom of column 27, and there is also a change of sheet between columns 27 and 28, i.e. two sheets are sewn together at this point. In the second half of the scroll the orthography and morphology of the Hebrew is different and there are spaces left blank.

The script type is called Hasmonaean in the field, the style of writing is formal, and the manuscript is traditionally dated to the late 2nd century BCE.

In the very early years of Dead Sea Scrolls research, scholars perceived an almost uniform writing style throughout the manuscript of 1QIsa[a] [22, 23] yet also acknowledged that different scribes could have shared a similar writing style [24, 25] but these initial observations were not much followed up. While only [5, 26] have stated that two scribes were each responsible for copying half of the manuscript, columns 1–27 and columns 28–54, most scholars have argued or assumed that the entire manuscript was copied by one scribe, with minor interventions by other, contemporaneous and also much later, scribes [27–29], and that orthographical and morphological differences between the two halves should be explained otherwise, for example, by assuming that two separate and dissimilar *Vorlagen* were used or that the *Vorlage* for the second half was a damaged manuscript [30–38].

No one, however, has provided detailed palaeographic arguments for writer identity or difference in 1QIsa[a], except for [28] who provided a palaeographic chart to argue for one main scribe. But the palaeographic chart in [28] is insufficient to demonstrate this for at least three reasons (additional details about the supposed scribal idiosyncrasies are provided in the S1 Supposed scribal idiosyncrasies in S1 File). Having been electronically produced it is unclear where, and how exactly, the characters were taken from. It is unclear whether "the typical form of the letters" is deemed typical because it is the most common form or because it is idiographic, understood as a subtle variation in graphic form that gives evidence of individuality [17]. Finally, the crucial question is how large amounts of data were processed to generate the chart. The number of instances of a specific Hebrew letter may run in the thousands in 1QIsa[a].

Here, pattern recognition and artificial intelligence techniques can assist researchers by processing large amounts of data and by producing quantitative analyses that are impossible for a human to perform. Over the years, within the field of pattern recognition, dedicated feature extraction techniques have been proposed and studied in identifying writers. By extracting useful quantitative data that is writer specific, these techniques are used on handwritten documents to produce feature vectors. In one of our earlier studies, we have tested both textural-based and grapheme-based features on a limited number of scrolls to identify scribes [39]. Textural-based features use the statistical information of slant and curvature of the handwritten characters. Grapheme-based features extract local structures of characters and then map them into a common space, similar to the so-called bag-of-words approach in text analysis [40].

We have already shown that extracting Hinge, a textural feature operating on the microlevel of handwriting, can be useful in identifying writers [41]. In the process of producing character shapes, writers subconsciously slow down and speed up their hand movements. For example, a bend within a character is an indication of where a slowing down took place, and the sharper the bend the greater the deceleration of the hand movement. Hinge uses this intuitive information between the static space and dynamic time to produce a feature vector.

Similar to the textural features, allographs (prototypical character shapes) can also be useful for writer identification [42]. Allographs can be obtained from either the full characters or from part/s of the characters. We have already worked with full characters and used them to create a codebook of the Dead Sea Scrolls characters for style development analysis [43].

The quantitative evidence is additional evidence that can stimulate palaeographers to explicate their qualitative analyses [21, 44]. Pattern recognition and artificial intelligence techniques do not give certainty of identification but they give statistical probabilities that can help the human expert understand and also decide between the likelihood of different possibilities.

The evidence from pattern recognition methods can be presented in numbers (quantification of distance; the choice of distance measures plays an important role) but also, more

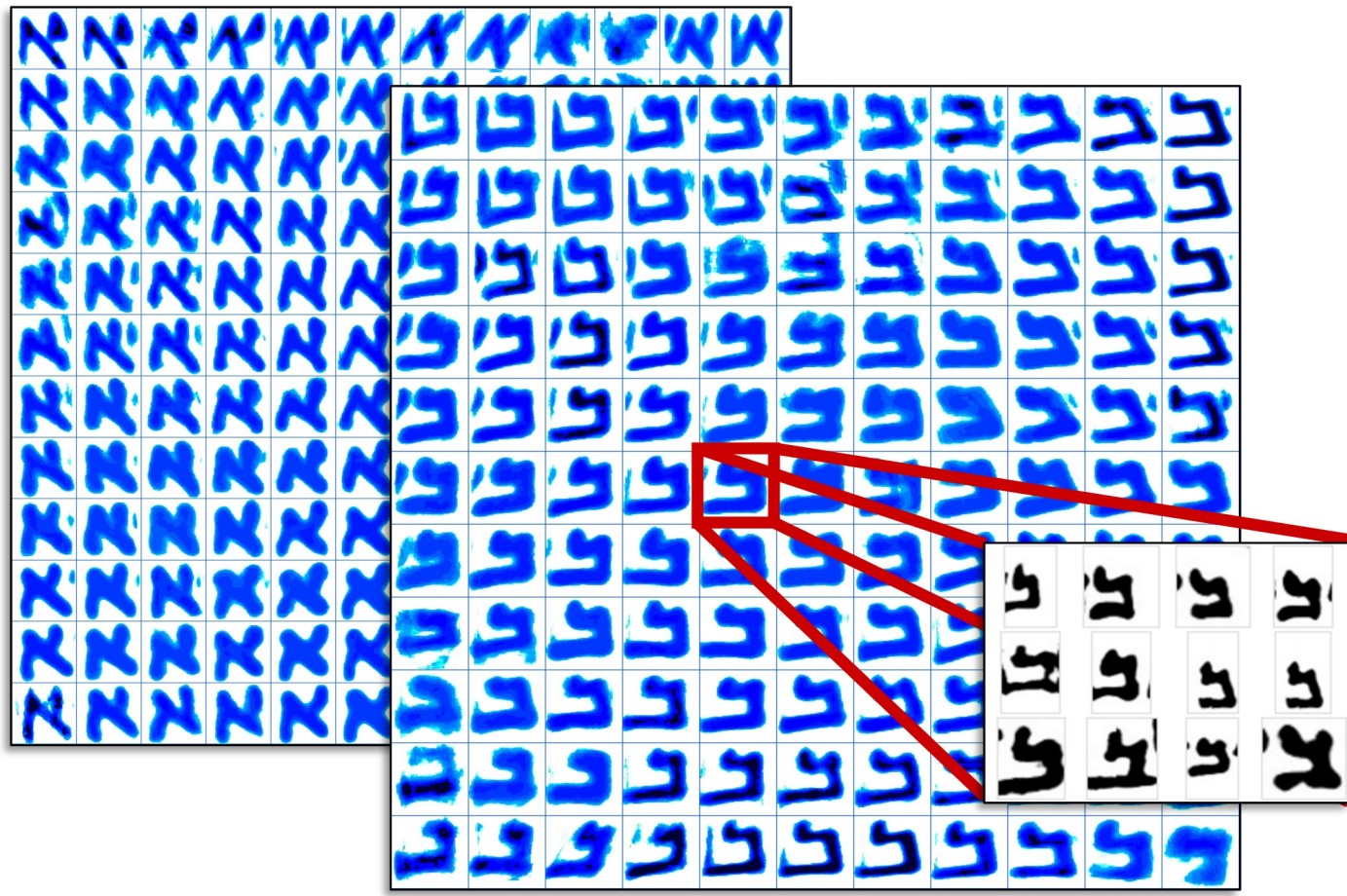

**Fig 1. Two 12x12 Kohonen maps (blue colourmaps) of (full) character *aleph* and *bet* from the DSS collection.** Each of the characters in the Kohonen maps are formed from multiple instances of similar characters (shown with zoomed box with red lines). These maps are useful for chronological style development analysis. In our current study of writer identification, Fraglets will be used instead of full character shapes to achieve more precise (robust) results.

helpfully, in two- or three-dimensional visualizations. Also, so-called Kohonen self-organizing feature maps (see Fig 1) and heatmaps may prove important for detecting a typical style of a letter (a centroid) that is the computed average of all particular instances that were most similar to it. Although such a centroid statistically is a reliable attractor for shapes that look like it, its visual pattern may not consist of a particular canonical or idealized form. Inspection of the individual instances belonging to a centroid (i.e., its members) will reveal the characteristics of that cluster of shapes. Such analyses may supplement exhaustive letter-by-letter analysis.

In terms of palaeography, we have found a new means to move the issue of writer identification in the Dead Sea Scrolls forward and present new evidence for two scribes. Our research demonstrates that two main scribes can be identified in 1QIsa[a] and that they coincide with columns 1–27 and columns 28–54. This study illustrates the advantage of using cutting-edge pattern recognition and artificial intelligence techniques for writer identification in the Dead Sea Scrolls when dealing with an almost uniform writing style that makes it difficult, if not near impossible, for researchers to assess writer identity or difference. Moreover, we show that procedures for cross-examination [17, 21] and falsification are in place by statistical and post-hoc visual analyses. Bridging artificial intelligence and traditional palaeography, our post-hoc visual analyses go beyond the state of the art by correlating the quantitative analyses to a level

suitable for researchers to be able to see what the computer 'sees', enabling a new way of looking at palaeographic evidence. Also, our analysis is *fully automatic*. We have no need to apply a semi-automatic first step of character reconstruction as in [45–47] that aim to imitate the ancient reed pen's movement, although it seems more likely that the stiff-flexible fibrous tip of the sea-rush stem must have been used, like in Egypt [48]. We have developed robust and sufficiently delicate binarization and extraction methods and have succeeded at extracting the ancient ink traces as they appear on digital images [49]. This is important because the ancient ink traces relate directly to a person's muscle movement and are person specific. For writer identification one should ideally work with the original written content only. The pattern recognition and artificial intelligence techniques should therefore be capable of focusing on the original written content only. Neither should it depend on modern character reconstructions.

In a way that was not possible before, our approach opens access to the tangible evidence of the hitherto almost completely inaccessible microlevel of the individual scribes of the Dead Sea Scrolls and the possibility to examine the different compositions copied by each of the scribes. The change of scribal hands in a literary manuscript or the identification of one and the same scribe in multiple manuscripts can be used as evidence to understand various forms of scribal collaboration that otherwise remain unknown to us. The number of literary manuscripts on which a scribe worked, either alone or with others, can serve as tangible evidence for understanding processes of textual and literary creation, circulation, and consumption. Together with other features such as content and genre, language and script, such clusters of literary manuscripts can contribute to scribal profiles of the anonymous scribes of the Dead Sea Scrolls, which, in turn, can shed new light on ancient Jewish scribal culture, in Hebrew and Aramaic, in the Graeco-Roman period. Here, we first tackle the palaeographic identification of these unknown scribes.

## 2 Materials and methods

In this section, we provide descriptions of:

- the dataset and the image preprocessing techniques(2.1),

- the primary analysis for textural features using pattern recognition techniques, for allographic features using artificial neural networks and a combination thereof (2.2),

- the second-level analysis using a different shape feature and performing statistical evaluation of the quality of the primary analysis (2.3), and

- the third-level post-hoc visual analysis (2.4).

For the choice of machine-learning methods ('AI'), we use deep learning at the level of image processing for binarization but deliberately avoid the extensive use of parameter-dense methods for the classification stage. It is difficult to reliably apply a deep-learning-based classification method to the given, limited data. The use of neural networks that are pretrained on the needed large collection of extraneous manuscripts ('transfer learning') would yield a severe problem in terms of transparency and explainability of results. The idea is to let the given data speak, using proven codebook methods (Kohonen maps: a type of artificial neural network) and proven feature methods designed explicitly for handwriting-style description. For the final decision making, traditional statistical tools are used.

Also, note that while the reading order in the Hebrew of 1QIsa[a] is from right to left, meaning that columns 1–27 are to the right and columns 28–54 to the left, instead in our machine-learning and statistical analyses the separate columns are ordered from left to right, so that,

e.g., the left-vs-right neighbours of a given column in Fraglet-shape space is the other way around from how one would read the columns in Hebrew.

Additional details and descriptions can be found in the S1 File.

## 2.1 Dataset and image preparation

In this study, we have used digital images of 1QIsaᵃ kindly provided to us by Brill Publishers [50]. There are 2463 images in the Brill scrolls collection with varied resolutions from 600 by 600 pixels to 2800 by 3400 pixels, approximately. For 1QIsaᵃ, we have images for columns 1–54 except for columns 16 and 46 (instead, columns 15 and 47 appear twice in the Brill collection; see Table 1 in S2 Image information of S1 File). The list of scan numbers and their corresponding column numbers are attached in the S2 Image information in S1 File. For the second-level analysis, we have also used the most recent digitized multi-spectral images of the Dead Sea Scrolls, kindly provided to us by the Israel Antiquities Authority (IAA); these images are also accessible on their Leon Levy Dead Sea Scrolls Digital Library website [51]. Although the IAA images do not contain any newly digitized version of 1QIsaᵃ, we have used this vast collection to extract dominant character shapes and produce self-organizing feature maps (see section 2.3).

The images of 1QIsaᵃ pass through multiple preprocessing measures to become suitable for pattern recognition-based techniques. Our first step in preprocessing is the image-binarization technique. In order to prevent any classification of the text-column images on the basis of irrelevant background patterns, a thorough binarization technique (BiNet) was applied, keeping the original ink traces intact [49]. After performing the binarization, the images were cleaned further by removing the adjacent columns that partially appear on the target columns' images. Finally, few minor affine transformations and stretching corrections were performed in a restrictive manner. These corrections are also targeted for aligning the texts where the text lines get twisted due to the degradation of the leather writing surface (see Fig 2). A more detailed explanation of image preparation can be found in the S3.1 Preprocessing: Binarization and alignment correction in S1 File.

Finally, to incorporate a realistic variation within a writer and check the system's robustness, we add noise to the data by applying random elastic 'rubber-sheet' transforms. The transforms produce augmented morphed data, which we use in the same system to check and compare changes in outcome with original unmorphed data (for more details, see S3.1.1 Image morphing: Adding random noise to the data in S1 File).

## 2.2 Primary analyses: Feature-space explorations

In order to represent the handwriting of 1QIsaᵃ, we applied feature extraction methods on the binarized cleaned images to translate the handwriting style into feature vectors. The data relates directly to the tangible evidence of the ink traces in the scrolls, ink penned by scribes. As writing is a moving process that involves muscle movements of the hand and arm it is determined by the rules of physics and can therefore be quantified.

Our feature extraction methods correlate the ink traces with the hands of the scribes on multiple levels. The allograph level of the whole character shape is easier to communicate to an audience, whereas the micro-level of textural features, such as Hinge, stands further away from the traditional visualization in the form of a palaeographic chart showing the whole character shape. Nonetheless, all these levels are equally directly related to the writing activity of ancient scribal hands that penned the ink on the scrolls.

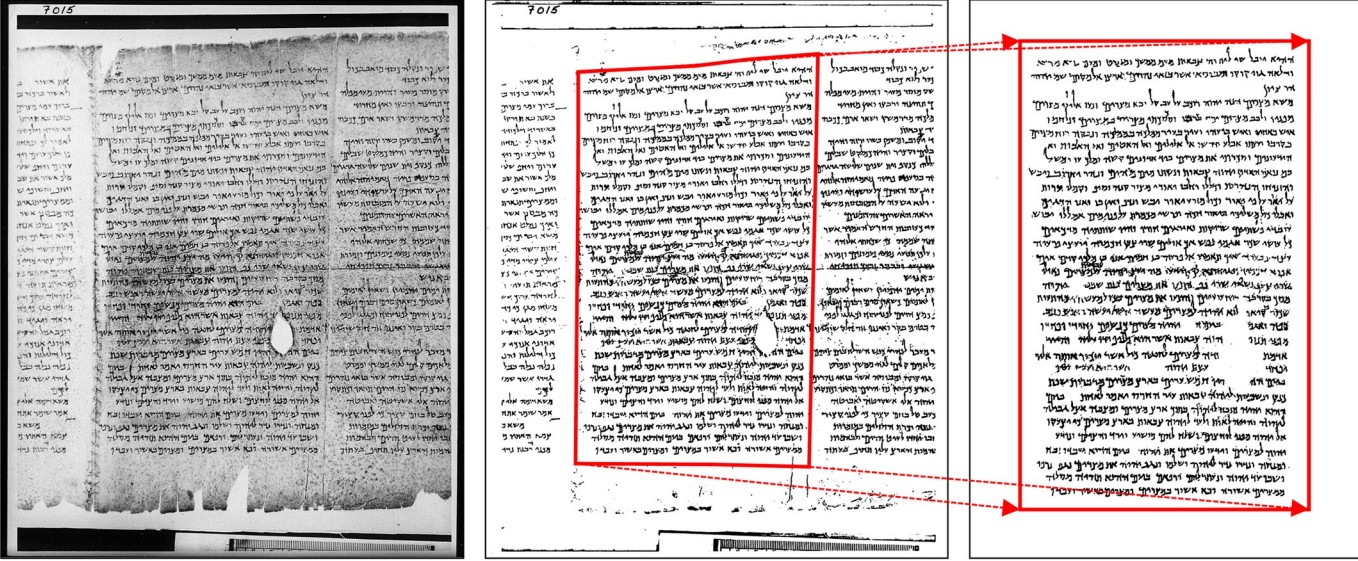

**Fig 2. (*from left to right*) Greyscale image of column 15, the corresponding binarized image using BiNet, and the cleaned-corrected image.** From the red boxes of the last two images one can see how the rotation and the geometric transformation is corrected to yield a better image for further processing. Reprinted from [50] under a CC BY license, with permission from Brill Publishers, original copyright 1995.

The question regarding 1QIsa[a] whether there are different scribes or one scribe was communicated to the researcher performing the primary analysis but no further information about the state of the art regarding this question in scrolls studies (see section 1) was communicated.

The primary analysis involved three steps.

**Step 1**. We have used three types of feature extraction techniques (detailed descriptions can be found in the S3.1.2 Feature extraction: Texture-level, S3.1.3 Feature extraction: Allograph level with neural networks, and S3.1.4 Adjoined Feature in S1 File):

- Textural feature extraction using pattern recognition techniques

- Allographic feature extraction using artificial neural networks

- Adjoined feature (a weighted combination of both textural and allographic features)

**Step 2**. After extracting features from each of the column images, we measured the distance between the feature files using the chi-square distance. The chi-square distance $d(x, y)$ is the distance between two histograms, namely $x = [x_1, .., x_n]$ and $y = [y_1, . . ., y_n]$, both having $n$ number of bins. In our case, the histograms are the feature vectors. During the calculation, we normalize the histograms, i.e. their entries sum up to one. The name of the distance is derived from Pearson's chi-square test statistics and the distance is defined as:

$$d(x, y) = \frac{1}{2} \sum \frac{(x_i - y_i)^2}{x_i + y_i} \qquad (1)$$

These distance files contain numbers which are relatively difficult to analyze without any reference distance. To solve this issue, we first move to clustering-techniques and then to probability curves. While clustering, we reduce the feature space into a three-dimensional space to facilitate the visualization of the feature vectors.

**Step 3**. A feature extraction method such as Hinge provides us with a large feature vector, containing hundreds of variables. Some features in the feature vector might not have a large

influence on the result. Therefore, the dimensionality of the data can be reduced in such a way that the most important aspects of the data remain. One way to do this is using Principal Component Analysis (PCA). It transforms the data into $n$ components that are independent of each other. Using PCA we go from multidimensionality to a three-dimensional space, and then inspect this three-dimensional plot to see if there is any significant movement of the point cloud.

In order to facilitate the decision-making process directly from the distance files (from step 2), one typical approach is to analyse probability curves; a False Acceptance Rate (FAR) curve (the likelihood that the system will incorrectly *accept* a writer) and a False Reject Rate (FRR) curve (the likelihood that the system will incorrectly *reject* a writer). These curves are generated from a known set of writers to incorporate all the variabilities. Depending on the distance between two feature vectors, the probability of being the same or a different writer can be determined. Unfortunately, in the Dead Sea Scrolls collection, there is *no certain* identification of known writers. In this study, we have avoided to introduce into our algorithm any *assumptions* by palaeographers about scribal identity or difference in the scrolls in general or in 1QIsa<sup>a</sup> specifically. This procedure ensures the outcome of this study to be independent from any bias.

Instead of being able to use probability curves, robust alternative techniques are needed for the Dead Sea Scrolls. In order to cross-check and test the quality of our findings from the primary analysis, we have used statistical evaluation as second-level analysis.

### 2.3 Secondary analyses: Statistical evaluations

The second-level analyses' goal is to independently assess whether there is a transition of style in the sequence of columns. The suspicion that there is a transition in the series of columns was communicated to the researcher performing this cross-check. However, until step 5, no more specific information was given about the sequence of columns where a style transition was observed in the primary analysis. The logistic tests performed in this part of the study were not influenced by any column information. This procedure ensures the independence of the second-level cross-examination.

The second-level analysis involved five steps; more detailed descriptions can be found in the S4 Secondary analyses in S1 File.

**Step 1**. In order to use a shape feature that is very different from those used in the primary phase of the study, it was proposed to use a fraglet approach, the so-called fragmented-connected component contours (fco3) [41, 52, 53]. In comparison to textural features that are concentrated around micro-details along the ink trace, fraglets contain more allographic information that may be understandable to a paleographer.

**Step 2**. A large Kohonen self-organizing feature map (SOFM) was computed, containing $80 \times 80$ centroids for such fraglets from the total IAA multi-spectral images collection that is at our disposal, yielding 6400 prototypical fraglets. About $600k$ randomly selected fraglets were used for this stage. Each centroid is based on about 94 fraglet instances. The use of the Kohonen map is not essential. Other clustering methods can be used; this step is not critical. But the Kohonen map has the advantage that the centroids that end up in the map change gradually, as opposed to a haphazard result of the ordering of centroids in, e.g., a k-means algorithm.

**Step 3**. For the series of columns, a histogram was computed for split-scan samples *a* and *b*, separately. In digital paleography and forensic handwriting this approach is used in order to check a reasonable response of the algorithm. It is expected that version *a* and *b* of a column of text should be close neighbours, under the assumption that a column was produced by a single scribe. If a hit list of neighbours for a query *a* of a column does not return the corresponding *b*

version in the top of the hit list of a search operation, results should be judged critically. Conversely, if the corresponding sample *b* appears at the top, the neighbouring hits will also have a larger probability of being produced by the same scribe [54].

**Step 4**. For each sample, the nearest neighbours were computed in the rest of the list. Bookkeeping was performed on the distance in feature space and the column number of the hits that were found.

**Step 5**. From the computed data (from steps 1-4), i.e., the distances and the column numbers of the nearest neighbour samples, four follow-up steps can be taken that help to determine whether the handwriting style is uniform throughout the manuscript of 1QIsa[a] or whether there are style differences.

**5a**. For testing the deviation of a random *voting pattern* for left-vs-right neighbours of a given column in fraglet-shape space, a Chi-square test was used. If there is a single signal source (scribe), nearest neighbours will fall to the left or right of a column in the series in a random pattern.

**5b**. A one-way analysis of variance, a t-test, was performed on the *distance values* of the left versus right nearest-neighbour matches in the series of columns.

**5c**. Apart from the distance between columns in fraglet-shape space, it is interesting to check the *estimated position* of a best-matching neighbour column for any given column in the series. If there is a single scribe, the nearest neighbour would appear in any column in the scroll. Conversely, if there are two scribes, the columns on the left would tend to have their best-matching neighbours on the left, and vice versa.

**5d**. If there is a phase transition in the sequence of columns, fitting a logistic curve on the variable 'average neighbour position' over columns should reveal the switching point reliably, i.e., with a high Pearson correlation of the fit. The number of the critical phase-transition column is the output of this test.

## 2.4 Tertiary analyses: Post-hoc visual analyses

The aim of the post-hoc visual analyses was to attempt to correlate the quantitative analyses from pattern recognition and artificial intelligence techniques with a qualitative analysis from a traditional palaeographic approach.

The third-level analysis involved three steps.

**Step 1**. For visual inspection by palaeographers, we created charts with full character shapes for individual Hebrew letters that can be found in the S5 Tertiary analyses in S1 File.

**Step 2**. In order to facilitate the complex process of visual inspection, we generated heatmaps for each character shape. The heatmaps are aggregated visualizations of the shape of each letter. These are made up of all particular instances of a letter and as such do not exist in one particular form. Thus, the use of heatmaps fulfils, through a sophisticated and robust procedure, the requirement from forensics to study each particular instance of a character. Also, the visualization by heatmaps may be an important step forward because they could work better than the palaeographic charts used traditionally in the field as they are not limited to one or more particular examples of which the indicative value can be doubtful but are made up of all instances of a letter.

**Step 3**. Suppose the primary analyses' results and the statistical tests in the second stage would turn out significant. In that case, a post-hoc visual analysis of the fraglet set contributing best to the discrimination between the left and right parts of the sequence is required to bridge the quantitative and the qualitative approaches. The fraglets refer to the characters' parts that can be more precise, distinctive, and informative in finding significant shape differences than the full characters. For each of the fraglet shapes, exploration can be performed to identify

their significance in separating two halves (if there exists a separation). Then, by running all possible combinations of 6400 fraglets and counting their presence in each image, a statistical view of the two halves can be obtained.

A dataset containing processed images (along with feature files and visualization script) is made available through Zenodo, an open-access repository [55].

## 3 Results

### 3.1 Primary analyses

Here we present the plots that result from the three types of feature extraction techniques that we used and the distance measurements between the feature files using the chi-square distance. The plots have been examined to find any possible clustering or any significant movement of the point cloud in the columns of 1QIsa[a]. We used the PCA technique on each of the feature collections and plotted them in a three-dimensional visual space (see Fig 3). Fig 3 shows the red points for each of the columns of 1QIsa[a].

In the next step, we used the colour red for columns 1–27 and the colour green for columns 28–54. Please note that this colouring works just as a label and has no effect/consequence on the experiments. The plots were then generated again for all the three types of features (see Figs 4, 6 and 7).

Fig 4 shows the plot using Hinge feature vectors on the full column images of 1QIsa[a]. There is a separation between the two sets of columns. Except for an outlier (column 29), the red and green points can be separated using a two-dimensional plane (similar to a piece of paper). This is visualized in Fig 5. The implication is that there might be a clear separation of the two sets of data, yet they are also close to each other.

As for column 29 appearing as an outlier in this part of the primary analysis, in the independent second-level analyses (see section 3.2), column 29 does not show up as a clear outlier. Also, in the primary analysis, column 29 is not an extreme outlier. Instead, it is close to the separation line of the two halves of the manuscript. Further tests can be performed in the future to conclude on a concrete reason for this.

Fig 6 shows the plot for the Fraglet feature from a 70 × 70 Kohonen SOFM. Here, the points are not that clearly separated as in the case of the Hinge feature. The reason might be because the Fraglet feature renders the physical shapes of characters, similar to what the human eyes see, it is less adequate (in this particular case) to determine any micro-level differences in the data.

In the last step, we combined both these features, Hinge and Fraglet. Fig 7 shows the plot for the combined feature, or Adjoined feature. A clear separation is visible here between the data points in the adjoined feature plot.

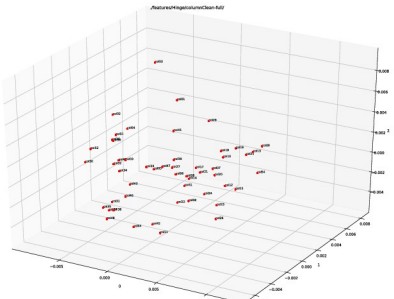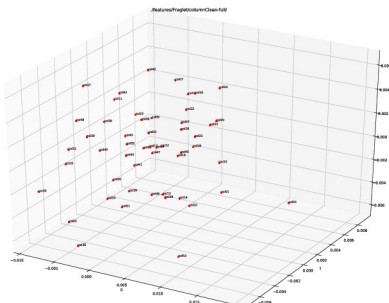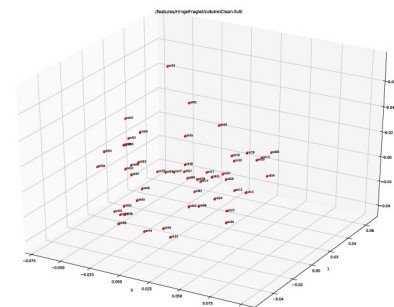

**Fig 3. Plots of different feature vectors in three-dimensional space using PCA (*from left to right*: Hinge, Fraglet, and Adjoined features).**

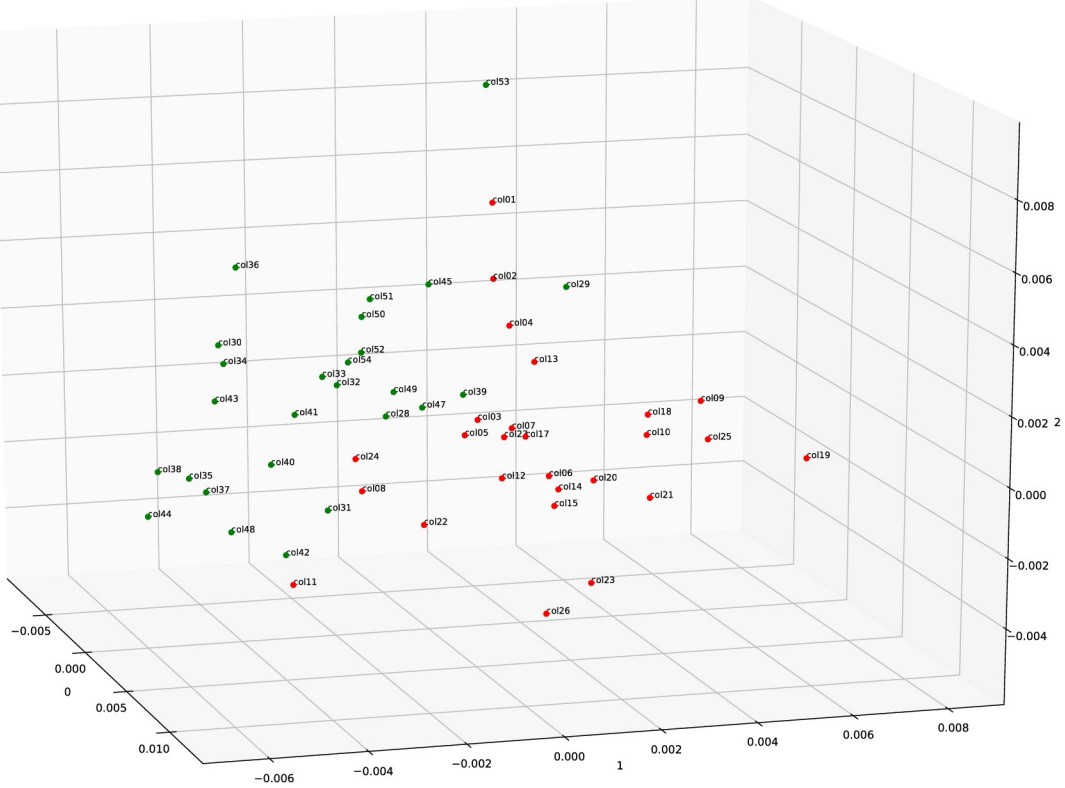

**Fig 4. Hinge feature plot (using PCA) of the full column images of 1QIsa[a] (red: Columns 1–27; green: Column 28–54).**

We also performed three random elastic 'rubber-sheet' transforms to the data with a displacement value of 1.0 and smoothing radius of 8.0 [56]. The elastic morphing is merely an addition of random noise to the handwritten data in a restrictive way (defined by the hyperparameters 1.0 and 8.0) so that it imitates the variability within a writer without damaging any originality of that writer. The transforms produce three augmented morphed images for each of the 1QIsa[a] columns (more details on the augmentation can be found in the S3.1.1 Image morphing: Adding random noise to the data in S1 File). Fig 8 shows the plot for augmented data. Again, even with the addition of noise, the plot shows a clear separation between the two halves.

Thus, the primary analysis indicates a significant difference between the two halves of 1QIsa[a] with a visibly clear separation in the point clouds of features.

### 3.2 Secondary analyses

Steps 1–4 (described in section 2.3) are pre-requisites to perform the tests in step 5. A detailed description of the first four steps can be found in the S4.1 Kohonen map of fragmented connected components in S1 File. It is important to note here that the fraglet features (fco3) used in the secondary analyses are derived from a different Kohonen SOFM than those used in the primary analyses (S3.1.3 Feature extraction: Allograph level with neural networks and S4.1

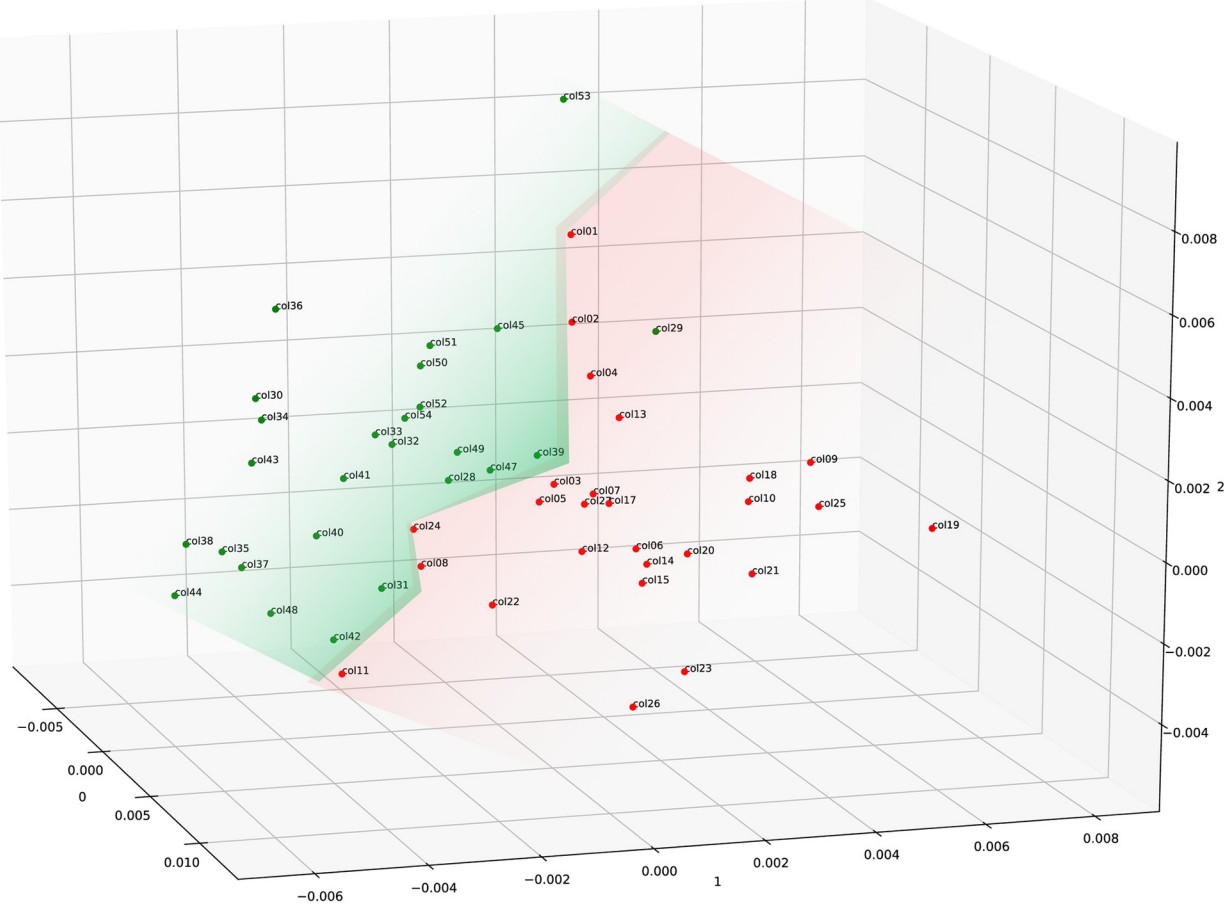

**Fig 5. Hinge feature plot (using PCA) of the full column images of 1QIsa<sup>a</sup> with a clear line of separation (red: Columns 1–27; green: Column 28–54).**

Kohonen map of fragmented connected components in S1 File). This is done to ensure the independence of two analyses and to perform cross-validation. The results of the statistical tests conducted in step 5 of second-level analyses are as follows.

**Step 5a**. Fig 9 shows the pattern of statistical probability that the left/right voting pattern deviates from random. A clear dip is present at the middle of the graph, confirming that at that point, the probability of nearest neighbours of a column falling to the left or right is very likely not an accident. This analysis, however, would be considered exploratory, and not a rigorous test, due to multiple testing over several time windows. Therefore, additional testing was done on the basis of the pattern of distances of columns to their nearest neighbours in shape space.

**Step 5b**. The average distance from query column to best match is 0.238 ($sd = 0.003$) on the left vs 0.231 ($sd = 0.008$) on the right, $p = 0.002$, which is significant at $\alpha = 0.005$. The inter-column distances are somewhat higher in the left series as compared to the right part, but their regularity is higher, given the lower standard deviation.

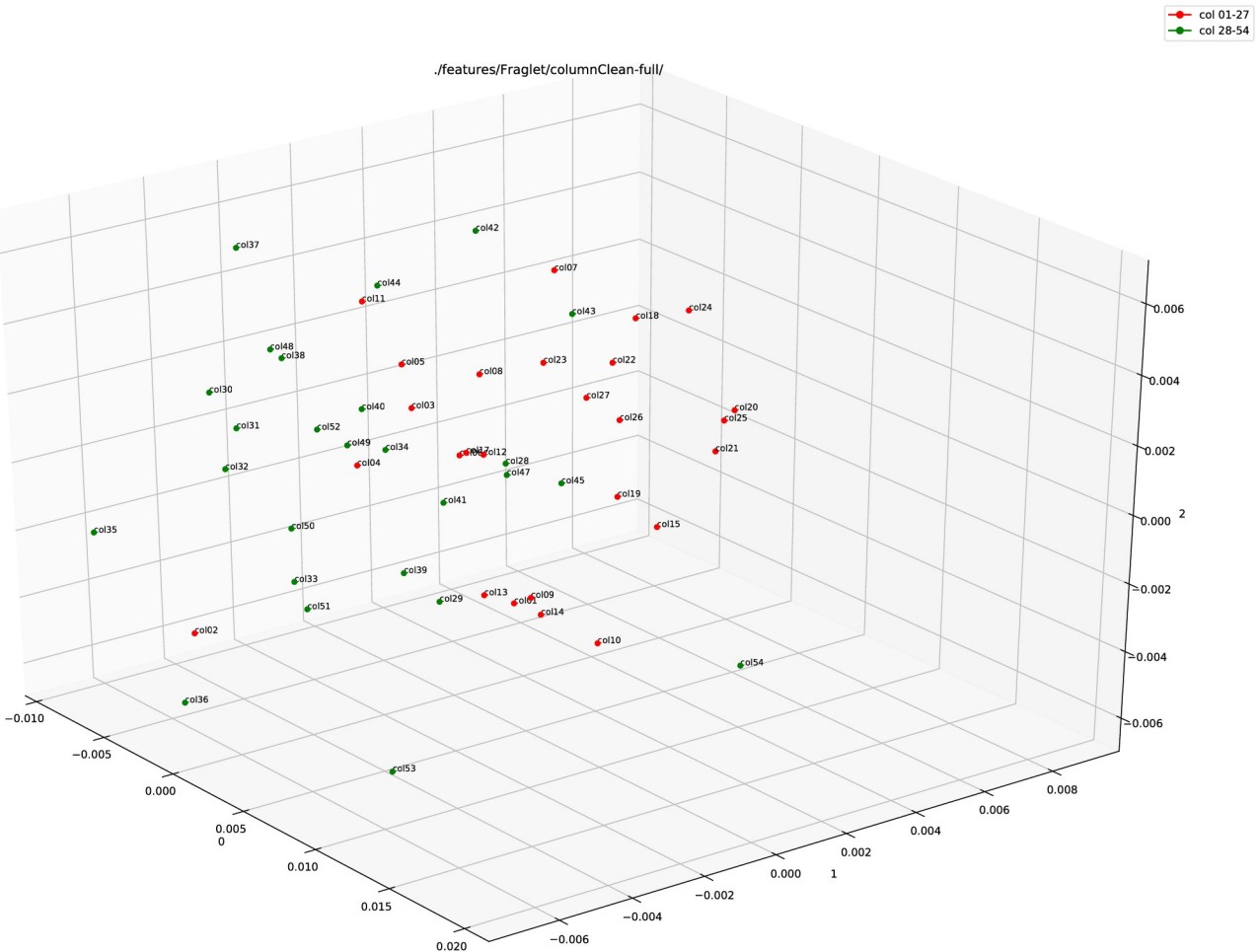

**Fig 6. Fraglet feature plot (using PCA) of the full column images of 1QIsa[a]** (red: Columns 1–27; green: Column 28–54).

This was further confirmed on the basis of an *F–test* for the statistical significance of a difference in variances (left vs right). The resulting *F–ratio* $var_{right}/var_{left}$ equals 1.78, $p = 0.04$, which is significant at $\alpha = 0.05$. The inter-column distances are therefore more variable in the right half of the series. This is indicative of more variable writing patterns in the second half of the manuscript.

**Step 5c**. Fig 10 shows the obtained column position of the best fitting neighbour for a column. Visually, from the smoothed curves, it can be seen that left of column 27, the average position of the hits is between column 20 through 25. On the right of column 27, the average position of hits is between column 30 and 35. A t-test indicates that the average nearest-neighbour column number for a column on the left is at column 24 ($sd = 4$), for a column on the right it is position 32 ($sd = 3.7$), where $p < 0.0001$ (see S4.2 Statistical tests on the fraglet feature distances in S1 File). Therefore the between-column similarity is highest 'ipsilateral' with respect to the cut point (column 27): '*left*' looks like left, '*right*' looks like right.

**Step 5d**. The results from steps 5a-5c are visually and statistically clear, but the valid question may be asked whether the actual point, i.e., the *column number of a phase transition* can also be computed? The logistic function or Fermi-Dirac function is usually used to model phase transitions in physics and biology. In the humanities, it can be used to model language

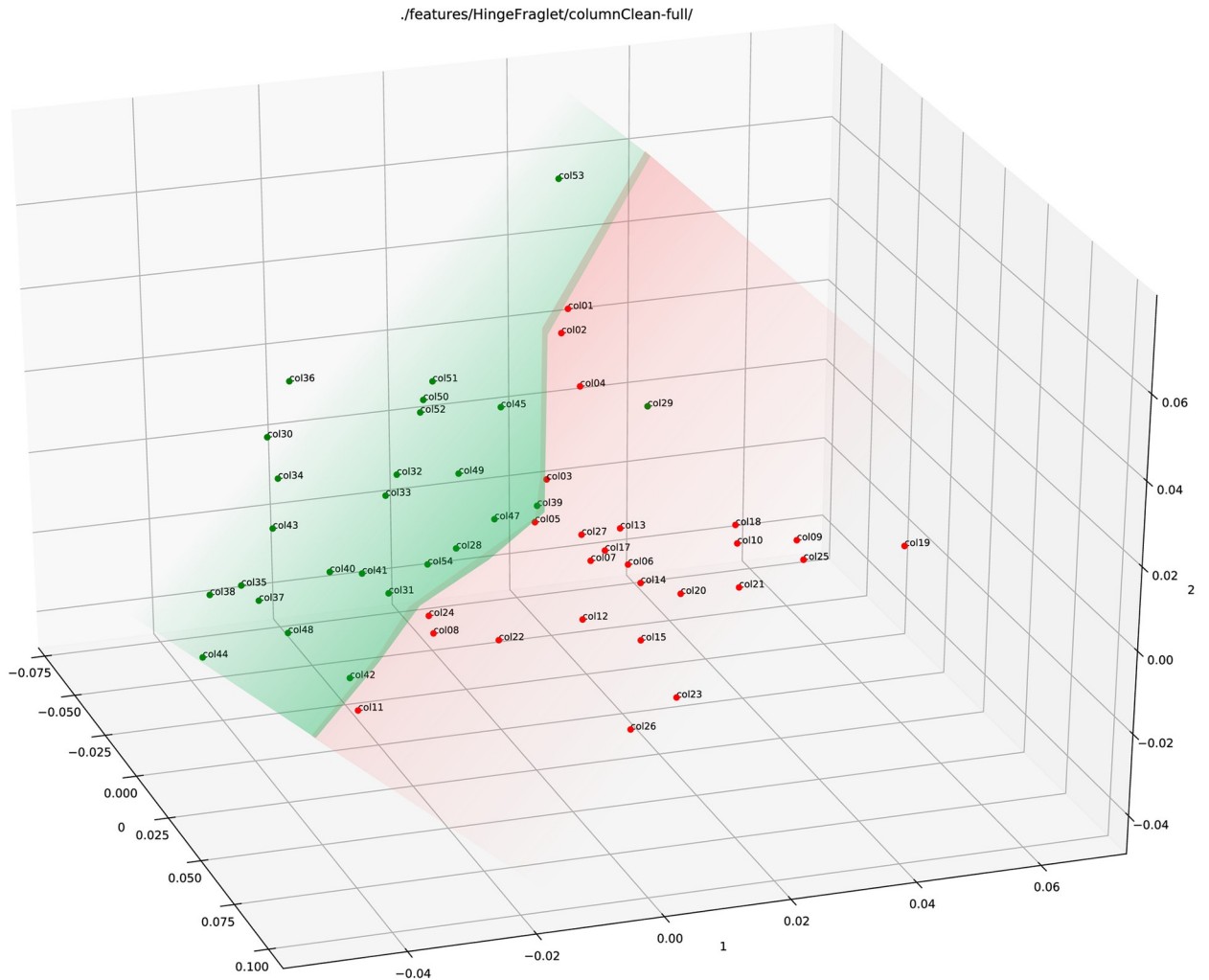

**Fig 7. Adjoined (Hinge+Fraglet) feature plot (using PCA) of the full column images of 1QIsa[a] (red: Columns 1–27; green: Column 28–54).**

change [57, 58]. Two types of analysis were performed to estimate the parameters of the logistic model:

$$f(x) = Y_{offset} + \frac{A}{1 + exp(b(x - x_{offset}))} \qquad (2)$$

where $x$ is the column number, $f(x)$ is the estimated average position of its nearest neighbour in the column series as measured in the fraglet shape space. Parameter $Y_{offset}$ represents the vertical offset, $A$ represents the scale factor, $b$ represents the steepness of the phase transition and $x_{offset}$ represents the column number where the phase transition occurs. In order to be very sure that a solution for the transition point is not haphazard, we will perform two very different estimation procedures for the logistic function. First, in order to allow a list of high-quality model fits to evolve without constraints, we used a Monte-Carlo estimation, randomly varying parameter values and remembering the best solutions. This sampling approach allows good results to emerge, without theoretical assumption. The second method is the more

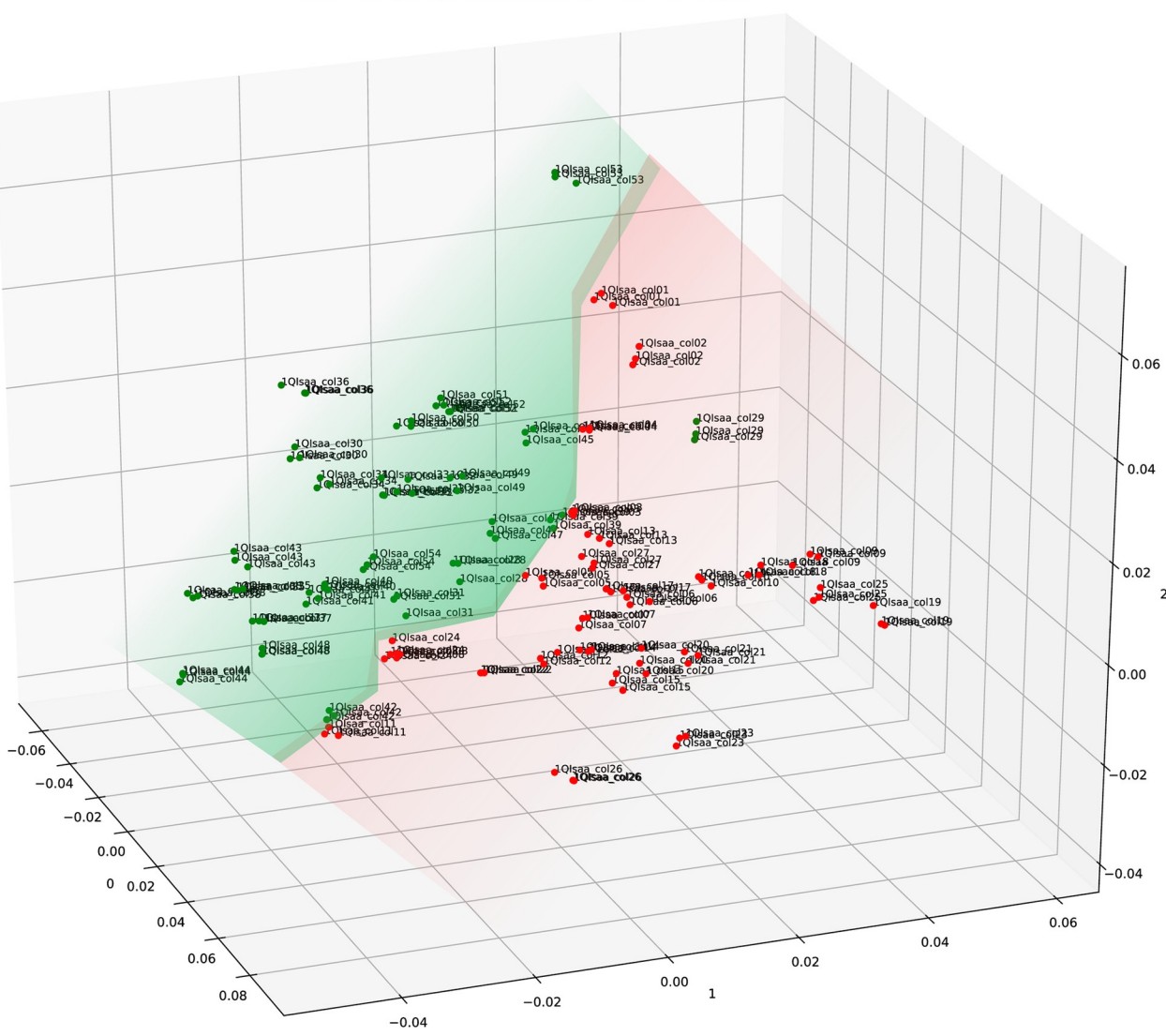

**Fig 8. Adjoined (Hinge+Fraglet) feature plot (using PCA) of the full column images of 1QIsa[a] (red: Columns 1–27; green: Column 28–54) with added noise (three random elastic 'rubber-sheet' transforms).**

traditional curve-fitting approach that uses the 'least-squares error' as the assumed constraint, to deliver a single best-effort solution. Without seeding a logistic function estimator with knowledge concerning the suspected column number 27, the output of the Monte-Carlo estimate can be found in Table 1.

In Table 1, the value of $x_{offset}$ means that the transition column is estimated to occur between column 27 and 28, with a transition steepness that is smooth lasting from column 24 to 32 (Fig 10). The fit of the sigmoid transition model is significant, with a correlation r that equals 0.74 ($p < 0.001$) on the raw data. An exact fit would have yielded r = 1.0. Although not perfect, r = 0.74 would be considered as a very robust correlation in empirical disciplines such as psychology and biology. The model would explain 55% of the variance in the data, which is

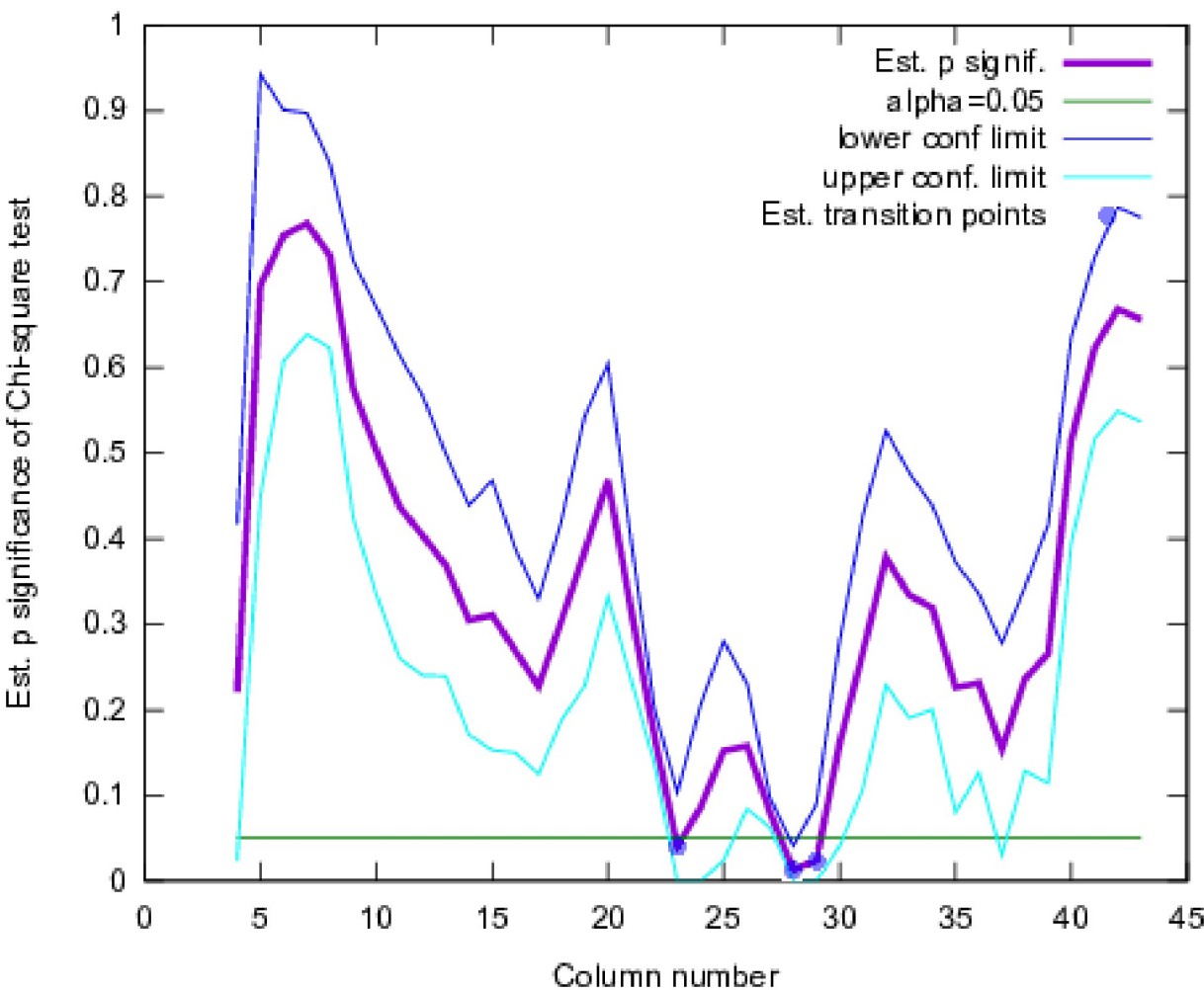

**Fig 9. Estimated significance probability for a left vs right chi-square test, averaged over a range of window sizes of 9-26.** With an alpha of 0.05, columns 23, 28, and 29 indicate that there is a discrepancy between the left and right neighbour votes. The general pattern suggests that something is changing in the statistics of the hit pattern left/right, around the middle of the column sequence.

not strange, given the fact that the logistic model is a stylized description of a time sequence with irregularities. If we smooth the irregularities over time, using a running average over a limited 3 or over 5 samples (columns), to smooth out the within writer variation, the correlation with the sigmoid increases considerably: if we smooth the column time series over 3 values, $r = 0.87$ (76% var. explained variance by sigmoid phase transition); if we smooth the column time series over 5 values, r = 0.93 (86% var. explained variance by sigmoid phase transition).

As a double check, the Monte Carlo-based fit was replicated with a more traditional least-squares curve fit (Python scipy package), yielding a phase transition at column 26.6 for raw data, with $r = 0.74$, at 26.2 for a smoothed time series with a window of three points ($r = 0.87$) and a transition at column 26 for a smoothed time series with a window of five points ($r = 0.94$). This double-check indicates that both traditional curve fitting and stochastic model fits yield a transition around the middle columns of 1QIsa[a]. Interestingly, also the quality of fit (correlation) is similar for these two very different estimation methods, adding to the trust in the found transition point.

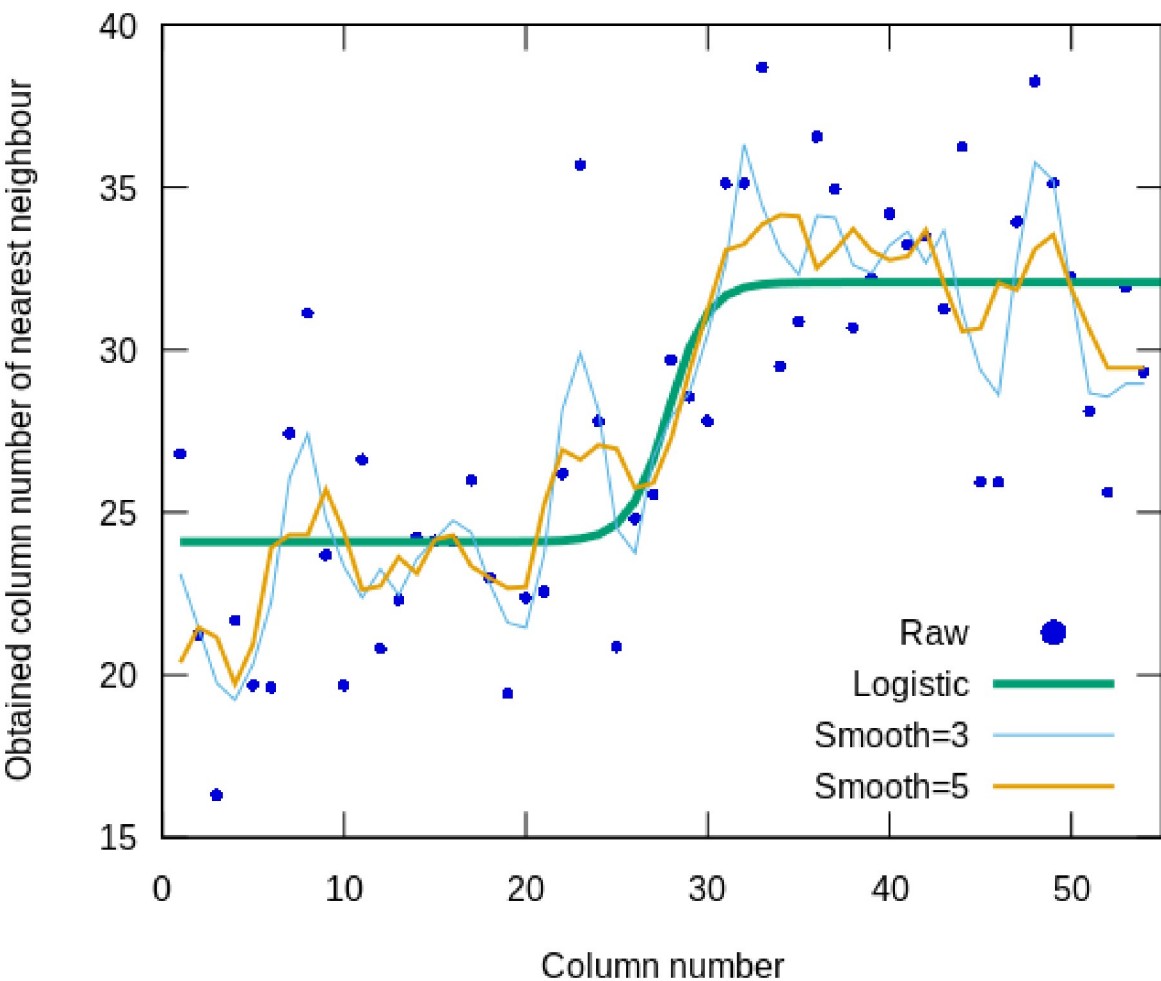

**Fig 10. Average serial column position of the nearest-neighbour of a column, measured in fraglet-feature space.** Raw samples (points), curves smoothed over 3 and over 5 points, and a best-fit logistic curve.

As a final check, a test was done to preclude any specificity of the results, by applying random elastic morphing to the image samples and fitting the logistic function on the average position of sample nearest neighbours in the column series. The original image samples 'ColA' and 'ColB' for a column were randomly morphed [56] into three variants each, yielding six perturbed versions of the original input image. Morphing parameters were chosen to obtain visible differences without affecting legibility (parameter values 1.0, 8.0). The logistic model estimation on the fraglet feature yielded very similar results compared to the raw unperturbed image samples: switching point at column 26.5 ($\sigma$ = 3.1), $\bar{r}$ = 0.82, using a 70x70 Kohonen

**Table 1. Resulting parameter values for a Monte-Carlo estimated logistic model ($r$ = 074).**

| parameter | value |
|---|---|
| $x_{offset}$ | 27.8 |
| $Y_{offset}$ | 24.1 |
| $A$ | 7.98 |
| b | 0.92 |

map and at column 28.0 ($\sigma = 1.6$), $\bar{r} = 0.74$, using an 80x80 Kohonen map of fraglets. In order to check whether the switching point is a particularity of the fraglet feature, the logistic model was also estimated for the Adjoined Hinge+Fraglet features, using the randomly morphed images. In this test, the switching point obtained was slightly more to the right, but still around the middle: 29.8 ($\sigma = 0.37$). This final check demonstrates that stress testing by introducing noise into the procedure confirms the robustness of our approach as well as validates our results.

Thus, the second-level analyses confirm the presence of two different clusters in writing style in a series of handwritten columns, to be called *left* and *right*. The confirmation occurs in three different ways:

- left/right votes for the relative serial position of the nearest neighbour of a column,

- distance to nearest neighbours on the left or right, and

- average serial column position of the nearest neighbour of a given column in fraglet-feature space.

The results from these analyses show that a transition point occurs at around column 27 to 29. The obtained logistic model fit in step 5d suggests that the transition is less sharp for the fraglet feature than for the feature combination, as evidenced from the standard deviation of the switch point.

### 3.3 Tertiary analyses

The results for our attempt to correlate by visualization the quantitative analyses from pattern recognition and artificial intelligence techniques to the level suitable for palaeographers to be able to see what the quantitative analyses 'see', in this case a clear separation in style, are as follows.

**Step 1**. Our charts with full character shapes for individual Hebrew letters improve significantly on the traditional palaeographic chart, such as in [28]. Each instance of a character can be directly traced back to its exact position in the manuscript of 1QIsa[a]. Also, there is no modern human hand involved, either in retracing the characters or in character reconstruction. The ink traces are extracted as is from the digital images and retain the movements once made by the ancient scribe's hand (see Fig 9 in S5 Tertiary analyses in S1 File).

However, as described in Section 1, due to the large number of characters from each column and the number of columns, the decision-making process from visual inspection alone of such charts may prove inadequate.

**Step 2**. A character heatmap is the normalized average character shape of individual letters extracted from the column images and aligned on their centroids (see Fig 11). The heatmaps are neither dependent nor produced from the primary and secondary analyses (subsections 3.1 and 3.2). They are entirely independent of pattern recognition and artificial intelligence-based tests. We present these heatmaps to produce an easy-to-use visualization for the palaeographers to observe any differences between letters coming from different columns.

We generated three different heatmaps for each letter, corresponding to the three aggregate levels for all columns of 1QIsa[a], for columns 1–27, and for columns 28–54 (for some examples, see Fig 12). Though the full-character shapes from Fig 12 seem to exhibit not that much differences among them, a close inspection reveals subtle differences between the two halves of 1QIsa[a]. These differences can be observed in the thickness of strokes and the positioning of connections between strokes. See, for example, the subtle difference in positioning of the left down stroke and the right upper stroke vis-à-vis the diagonal stroke of *aleph* and the slight

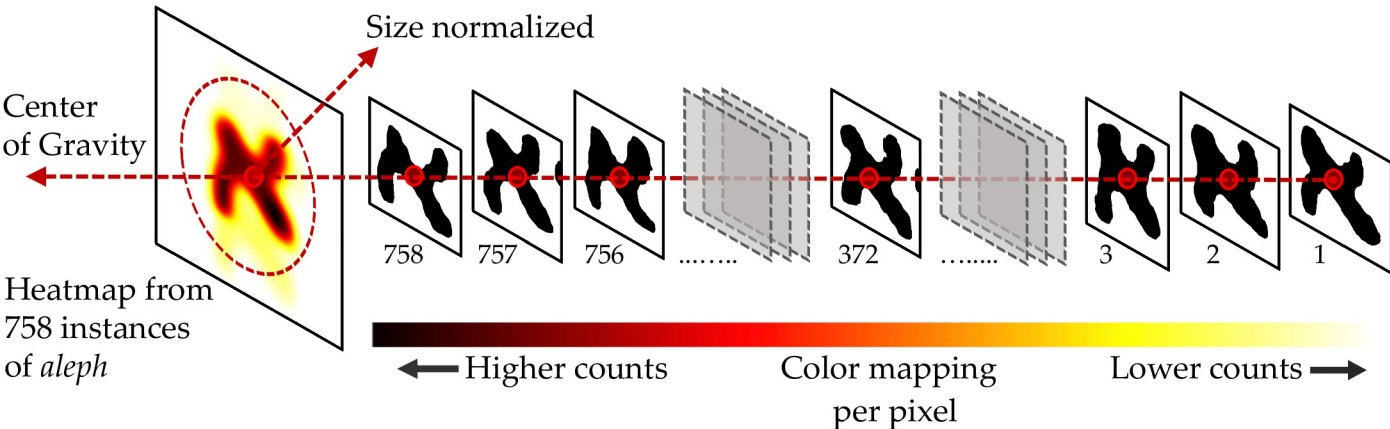

**Fig 11. An illustration of how heatmaps of normalized average character-shapes are generated for individual letters (example: *Aleph*).**

difference in thickness of the diagonal stroke, or the slight difference in thickness and length of the horizontal stroke of *resh* (see Fig 13).

In a traditional palaeographic chart such differences might be deemed insignificant and explicable as normal variations within the handwriting of one writer. If that were the case, i.e., what we see is normal within writer variability, then for 1QIsa$^a$ one would expect the same distribution of writing style across all columns, which is not the case. Rather, the primary analyses as well as the statistical tests (5a–5d) indicated a significant separation and a clear distribution of the two halves of the manuscript of 1QIsa$^a$ on either side of the divide.

Heatmaps should be inspected with a different understanding. Heatmaps are different from traditional palaeographic charts in that they represent the aggregated visualizations of the shape of each letter, hundreds per letter in the case of 1QIsa$^a$. Given the large number (count) of samples and the fact that the center position estimate is stable, then the remaining differences after averaging are an indication of an underlying structural difference. Thus, in heatmaps, the subtle differences we see between the different aggregate levels are indicative *if* the

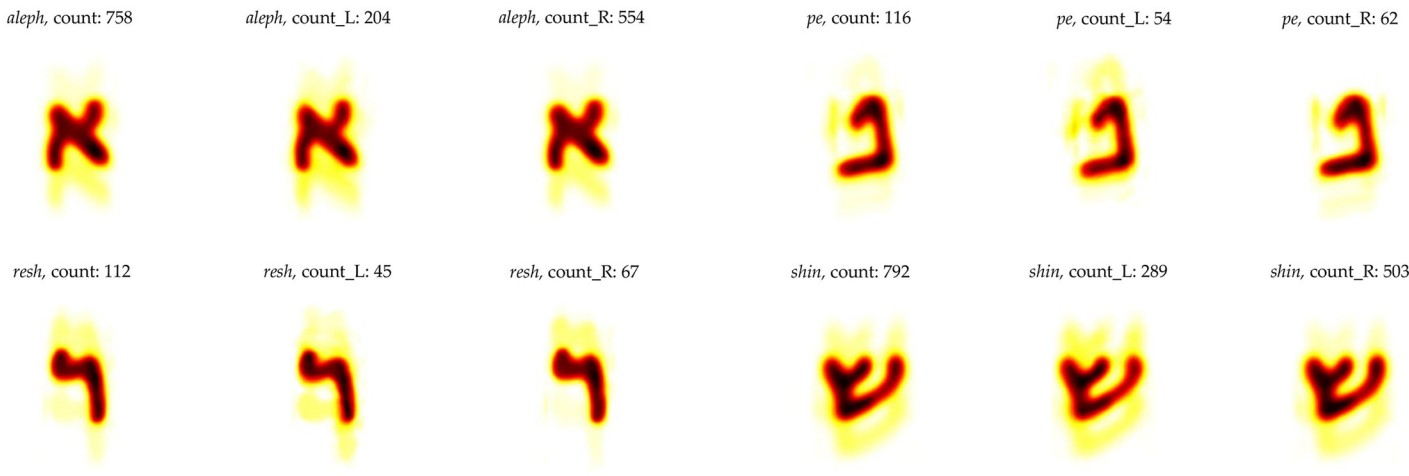

**Fig 12. Individual character heatmaps of *aleph*, *pe*, *resh*, and *shin* from 1QIsa$^a$.** On the top left, the first *aleph* is aggregated from all the columns, the next one is from columns 1–27, and the final one is from columns 28–54. The same applies for the other three characters.

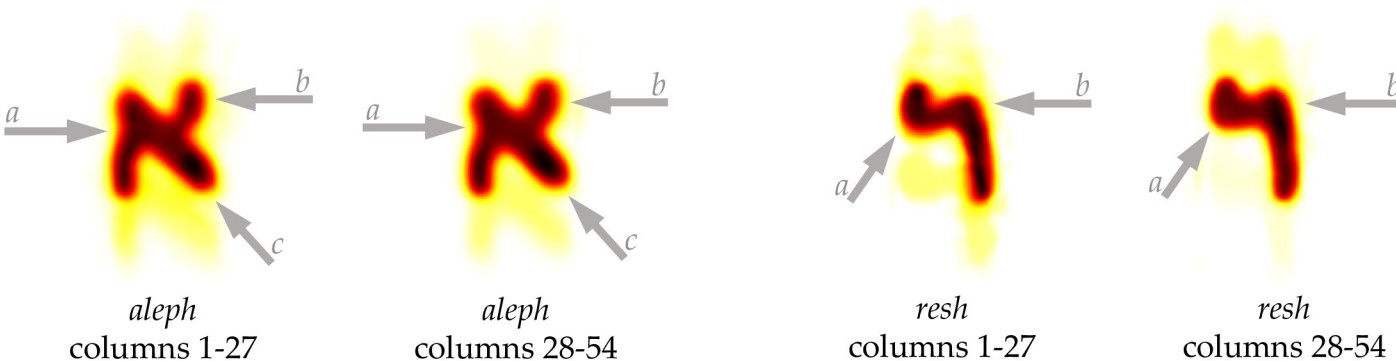

| *aleph* | *aleph* | *resh* | *resh* |
| columns 1-27 | columns 28-54 | columns 1-27 | columns 28-54 |

**Fig 13. A zoomed in view of *aleph* and *resh* from Fig 12.** In the case of *aleph*: the subtle difference in positioning of the left down stroke (*a*), the right upper stroke vis-à-vis the diagonal stroke (*b*), and the slight difference in thickness of the diagonal stroke (*c*). in the case of *resh*: the curvature of the top stroke (*a*), and the slight difference in thickness and length of the horizontal stroke (*b*).

separation between the different levels has also turned out significant otherwise, which is the case for 1QIsa[a].

Note, that we have only used the automatically recognized characters to generate the heatmaps from the columns of 1QIsa[a]. The number of generated *aleph*s for the heatmaps is 758, while the total number of *aleph*s in 1QIsa[a] is 5011. These 758 *aleph*s were automatically extracted by the computer on the basis of known shape structures, and the extracted characters come from all columns, representing a general distribution. This extraction is extremely efficient and has the advantage that it does not require human intervention. Our goal is not to produce an exhaustive enumeration of all *aleph*s in the manuscript, rather than produce heatmaps that cover all columns with a sufficient number of examples. Therefore, the heatmaps presented here are robust enough to indicate the differences (previous studies can also back this claim [59]). To demonstrate the robustness: with the current number of *aleph*s, any pixel of the heatmap with mid-intensity (here, orange with 0.5 intensity, band 0.46 to 0.54, and total intensity being $0 - 1$) has a probability of 0.05 for that one pixel to give different results. So even if we were to increase the number of instances of a particular character, the resulting heatmap will not change significantly (it is possible to request heatmaps from all the individual characters by emailing the corresponding author).

**Step 3**. After having found statistically significant differences in the neighbourhood structure for columns in the scroll, and after having confirmed that a transition occurs at about the middle of the column series, a more detailed analysis is warranted. Please note, that the actual evidence for the differences comes from the primary and secondary analyses, whereas the current focus is illustrative only. The statistical differences obtained are the result of many small textural and allographic differences. For these allographic differences it is also important to keep in mind that an exhaustive list of possible allographs is not required: an allographic codebook approach will work very well, if it is sufficiently diverse [52]. In the current problem, some of the allographs appear to be more different in their occurrence over the left and right columns, and we can take a look at them for illustrative purposes, while remembering that this concerns partial evidence from the extremes of the distribution.

For the fraglet feature, a selection was made of the most informative fraglets that are able to discriminate between the leftmost ($i <= 27$) and the rightmost columns ($i > 27$) in the series. Please note, that the number of fraglets in the SOFM is 6400. From these 6400 fraglets, we automatically generated sets of fraglets to visualize the differences between the two halves of the manuscript. Thus, we ran tests with thousands of combinations of fraglet sets, each

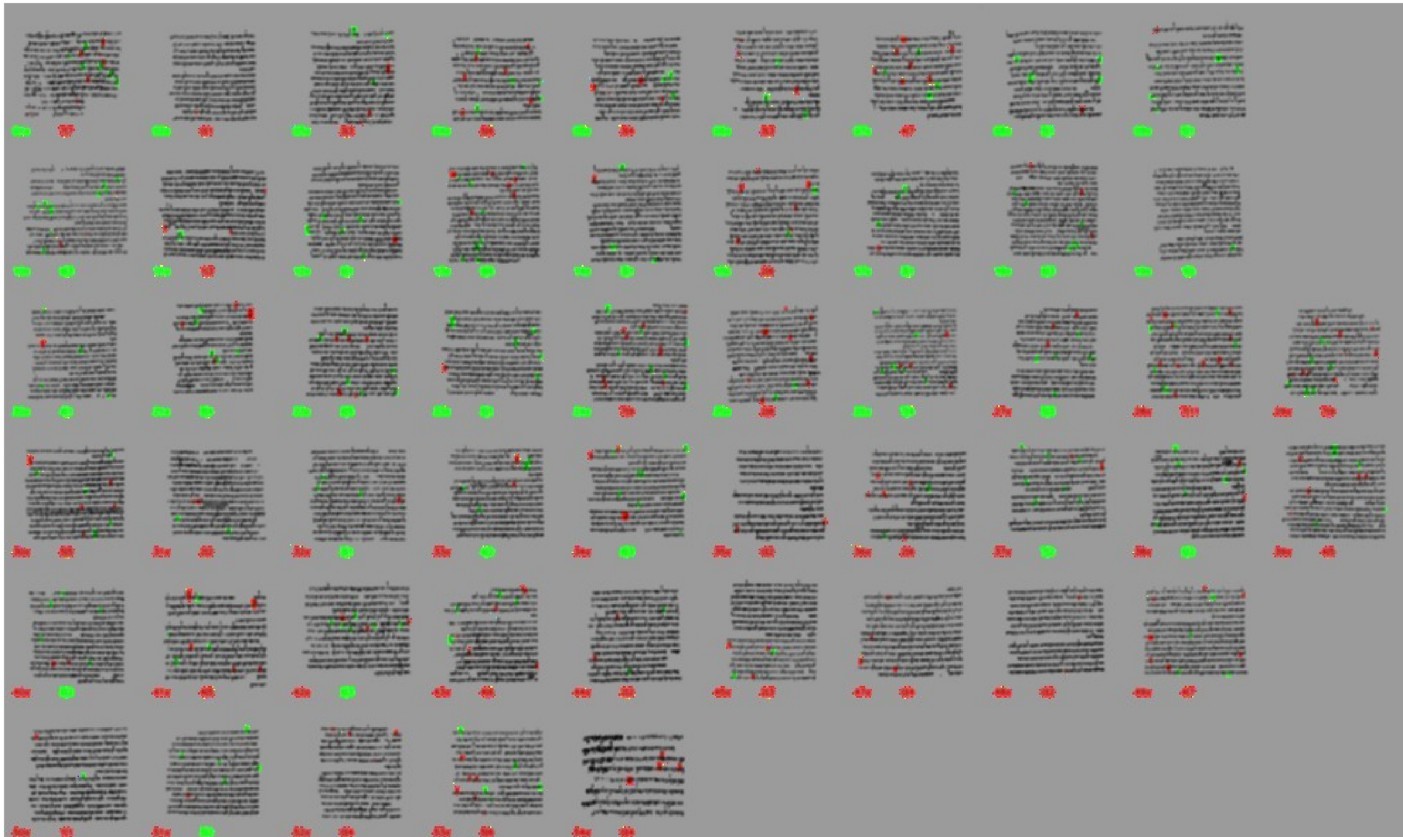

**Fig 14. Visually enhanced presence of typical 'left' fraglets (green) and 'right' fraglets, separately for the 'a' split scans (top-halves) of the columns.**

providing a new overview. Figs 14 and 15 show such an overview for the relevant columns (more images can be found in S5 Tertiary analyses, see Figs 10 and 11 in S1 File). Below each column thumbnail, the left blob indicates the ground truth ('left series' is green, 'right series' is red), whereas the colour immediately to the right of it shows the colour that the subset of most-informative fraglets predicts. These figures illustrate the statistical view of a separation between the two halves of the manuscript.

## 4 Discussion and conclusions

The aim of this study was to tackle the palaeographic identification of the unknown scribes of the Dead Sea Scrolls, exemplified by 1QIsa[a]. The question for 1QIsa[a] was whether subtle differences in writing should be regarded as normal variations in the handwriting of one scribe or as similar scripts of two different scribes and, if the latter, whether the writing of the two scribes coincides with the two halves of the manuscript. The evidence collection was presented in a chronological manner.

Firstly, an independent observation was made that in feature spaces, the left and right part of the column series, ended up in different regions. Several feature methods confirmed this observation. The preferred explanation is that there were two main scribes responsible for copying 1QIsa[a], their work indeed separated between columns 27 and 28 by a three-line lacuna at the bottom of column 27. We see that there is a clear separation between the data points in both the Hinge and the Adjoined feature plot (Figs 5 and 7). If we consider an explanation in

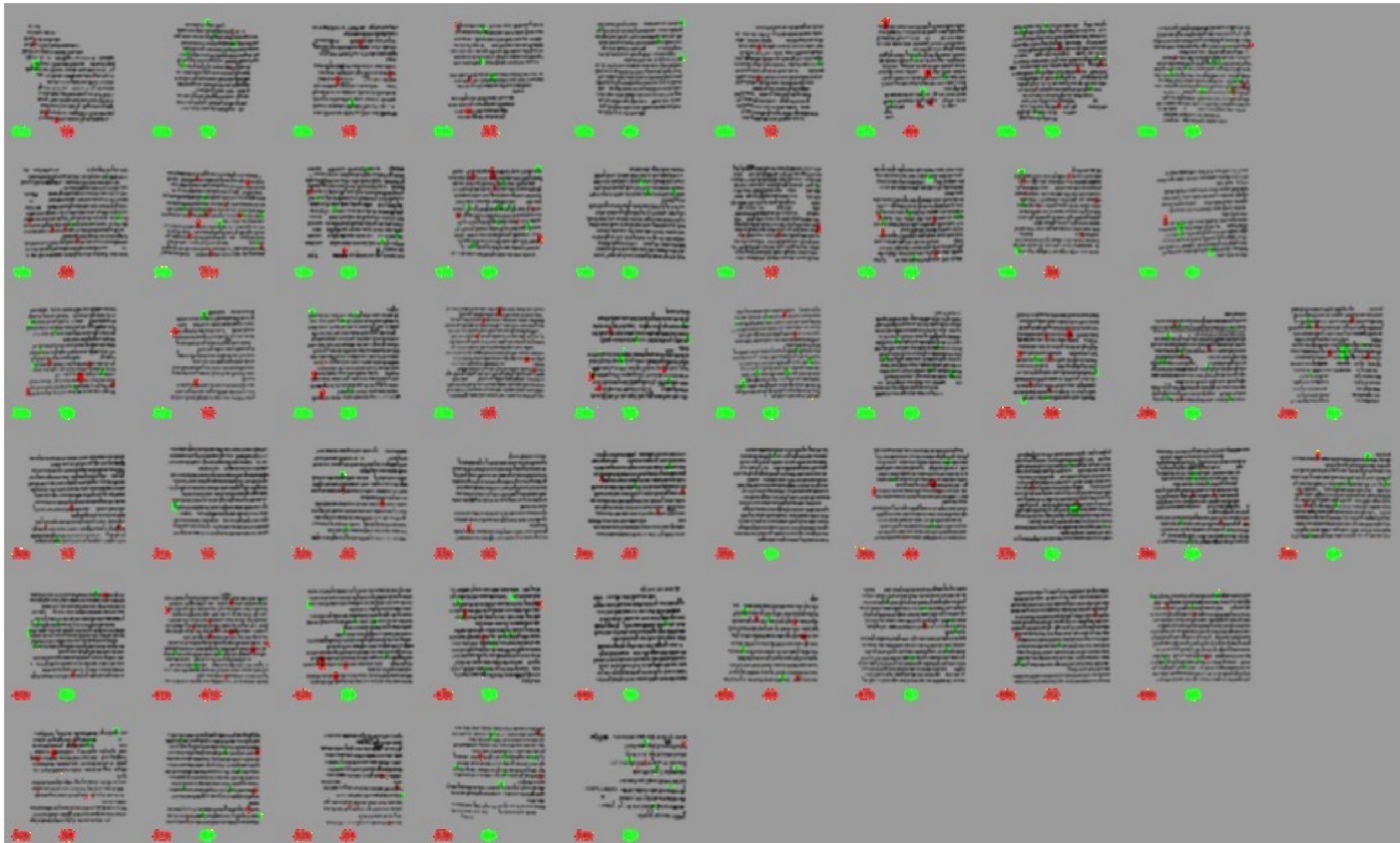

**Fig 15. Visually enhanced presence of typical 'left' fraglets (green) and 'right' fraglets, separately for the 'b' split scans (bottom-halves) of the columns.**

terms of a large variability within one single scribe, then the question remains why the points are not randomly scattered (between the two sets of columns) on the PCA space in the Adjoined feature plot. Instead, there is a clear indication of separation, at least from one of the angles of the plot space. Therefore, a more likely scenario is two different scribes working closely together and trying to keep the same style of writing yet revealing themselves, their individuality, in the textural feature space.

Secondly, a series of tests was performed on a separate shape feature, a Kohonen map of fragmented contours. A series of five questions was asked, starting with a statistical test whether the pattern of neighbours on the left or right of any given column deviates from the expected random pattern for the case of a single writing style. Because these tests clearly show that the neighbourhood structure is not random, additional analyses were warranted. The distances between columns, as measured in the Fraglet-usage space, also showed a highly significant pattern. We also demonstrated a difference in distance variances such that the variance is higher in the second part of the manuscript, which is indicative of more variable writing patterns. Finally, the serial column number for the nearest neighbour of each column shows a distinct transition at about the middle of the column series in the scroll. Fitting a logistic model delivered an estimate of the region where this transition occurs, i.e., around column number 27–29. This point is found without coercion, and emerges from two very different quantitative approaches (a least-squares and a separate Monte-Carlo analysis) on the time series of the column numbers of nearest neighbour matches, for each column. In simple terms: columns on the left clearly tend to yield nearest neighbours on the left, columns on the right clearly tend to

yield nearest neighbours on the right. This outcome was further confirmed by our stress test that introduced noise in the form of random elastic morphing: the results are insensitive to the noise introduced in the data, i.e. they stay the same, demonstrating the robustness of our approach and of our findings. The bistable configuration of writing styles is thus confirmed by an additional fit of the logistic model on the randomly morphed column images, in the original feature space (Fraglets) and also for the Hinge and the Adjoined (Hinge+Fraglets) feature vector. All analyses confirm the presence of a switch point. For Fraglets, the position of the switch point was confirmed, for the Hinge feature, it is estimated to occur a bit more to the right, but in any case with a high reliability (high $r$, low $\sigma$). Therefore, these secondary analyses confirm the suspicion raised on the basis of the exploratory primary analyses by other researchers in the team.

Thirdly, our fully-automatic generation of charts with full character shapes for individual Hebrew letters extracted from the digital images of the ancient manuscript of 1QIsa[a] greatly advances how palaeographic charts have been previously produced, while the subtle differences visible upon close inspection post-hoc of the heatmaps (both thickness and angular differences exist) also show that the use of heatmaps can help to bridge the quantitative analyses and traditional palaeography. Moreover, a post-hoc visual analysis on the most discriminative fraglets in the Kohonen 'bag of visual words', which is now allowable given the obtained statistical significance of differences between 'left' and 'right' in other measures, illustrates the transition point and the differential evidence by colour-marked fraglets in the column images. To be sure, the reverse is also true: if there were no statistical significance of differences between 'left' and 'right', then it would not have been allowable to look for evidence of difference in post-hoc visual analyses.

Yet, there are at least three variables that we need to be transparent about because these may affect the results, though not alter them significantly. These three variables are: material degradation, writing implements, ink deposition and writing conditions, and limitations on character extraction.

Regarding material degradation, we have to keep in mind that the scrolls, and by extension the images that constitute the data for our pattern recognition and artificial intelligence techniques, have degraded over the centuries and are not anymore in the shape they were once produced. This degradation causes an amount of uncertainty over the derived results, even though we tried our best to extract the original characters using state-of-the-art methods.

In general, writing implements and writing conditions can have significant impact on the outcome of the copied scrolls. The use of writing implements could differ in the cutting of the pen's nib and writing conditions could change in the course of time [7]. Although there is no evidence that different writing implements were used in 1QIsa[a] or a change in writing conditions occurred, the general point is that the specific writing implement or a change in writing conditions have an effect on the ink deposition, which in turn affects our modern extraction process of the original characters.

Finally, regarding limitations on extraction, note that character extraction can never be perfect. Nevertheless, we are confident with our methodology, and it clearly shows excellent extraction results, both qualitatively and quantitatively. Additionally, our feature extraction methods are tested on an independent dataset: 'Firemaker image collection for bench-marking forensic writer identification' [60]. Furthermore, the statistical tests are methodologically robust, independent of the data they are tested on, and further validated by stress testing that introduced noise.

The discussion of these variables is not to cast doubt on our study's outcome, which remains inherently sturdy, but reminds us that the techniques from pattern recognition and

artificial intelligence do not give certainty of identification but statistically proven probabilities that can help the human expert understand and decide between different possibilities.

Regardless of these variables, this research is by far the most comprehensive and elaborate study on writer identification on historical manuscripts using state-of-the-art computer-based techniques. The use of feature extractions on both macro- and microlevels of character shapes is extensive, gauging a writer's mimetic (cultural) and genetic (bio-mechanical) traits, respectively. The methods used here are rooted in earlier work in forensic writer identification [53, 54, 61, 62]. The minimal use of human interference, the cross-checks and re-validation through statistical tests and stress tests make this study unique and lay the foundation for future advanced studies. The conclusion is that the use of robust pattern recognition and artificial intelligence techniques is a breakthrough for the palaeography of writer identification in the Dead Sea Scrolls.

For 1QIsa[a] we have found new evidence for two separate clusters, with a clear break, more or less mid-point the manuscript, demonstrating, despite the near uniform handwriting, the presence of two writing styles of two different scribes in columns 1–27 and columns 28–54. While the differences between the two halves might seem small, in the sense that they lie very near each other, the individual points (columns) do not go into each other's areas and the break being statistically significant makes the separation a clear one.

With regard to the above-mentioned variable of writing implements and writing conditions, for 1QIsa[a] a change of pen, for example, is in itself not a sufficient explanation for the data and the statistical significance of the clear separation. This does not mean that a change of pen did not occur. There may very well have been a change of pens, with the change of scribes and also within one scribe sharpening the nib of the pen. The point is that the Hinge and Fraglet features independently tap into different information levels of the handwriting (Fraglets contain the larger, complicated character fragments, while the Hinge feature concerns local curvature) yet both methods point to a clear break in the data and separation of two clusters, which weakens the change-of-pen argument. Hinge is looking at the joint-angle distribution, which gets almost no impact from a change of pen (while stroke width does, but this is not what Hinge looks at). Even if a scribe changes pens or sharpens the nib of the pen, he is still limited or defined by his motor movement, which is what Hinge analyses. Fraglet looks at the contour shape (physical appearance) of the characters, which is also less impacted by differences in pen. Now, in our study, both these features independently confirm the same outcome, a statistically significant separation in the data so that there are two clear clusters. So even if there was a change in pen, these two features confirm the change in scribes.

Furthermore, the two scribes show different writing patterns: we have demonstrated, on the basis of variance of the Fraglet distances, that the second scribe shows more variable writing patterns.

Although one cannot rule out completely that the clear separation between the two halves of the manuscript and the difference in writing patterns are due to a change of writing implement (a different pen), writing fatigue or some injury that the writer suffered when moving on to the second half of the manuscript, the more straightforward explanation is that a change in scribes occurred. The presence of two scribes in 1QIsa[a] better explains the combined data concerning the fraglet and allographic levels of handwriting.

The similarity in handwriting between different scribes can indicate a common training shared by the scribes, perhaps in a school setting or otherwise close social setting, such as in a family context a father having taught a son to write. For five documentary texts it has been suggested that the similarity in script may be the result from a common school training [3]. We have otherwise no concrete evidence for such schools but their presence must be presumed [4, 63, 64]. Regardless of the exact explanation, our study demonstrates the ability to closely

mirror another scribe's writing style, so much so that modern scholars have not been able to distinguish between the two scribes of 1QIsa[a]. This mimetic ability may testify to a degree of scribal professionalism, despite modern researchers having characterized 1QIsa[a] as a sloppy manuscript, e.g., [65].

Furthermore, one of the crucial outcomes of our research is also the need for palaeographers in Dead Sea Scrolls studies to be aware that similarity in handwriting need not imply writer identity. Is it not strange that there are these very clear, statistically significant differences on the different levels of the handwriting in 1QIsa[a] and that this has not been noticed? Instead of asking whether traditional palaeography really captures everything, our study shows the need for and added value of collaboration between the disciplines. This may also apply to other ancient corpora that face similar palaeographic challenges, such as ancient Greek manuscripts [66, 67].

Our conclusion for 1QIsa[a] that there were two main scribes also sheds new light on the production of biblical manuscripts in ancient Judea. We have provided new, tangible evidence that such texts were not copied by a single scribe only but that multiple scribes, while carefully mirroring another scribe's writing style, could closely collaborate on one particular manuscript of a text that would come to be regarded and revered as biblical.

## Supporting information

**S1 File.**
(PDF)

## Acknowledgments

The authors owe a special debt of gratitude to Eibert Tigchelaar, Drew Longacre, Gemma Hayes, Jonathan Ben-Dov, Hugh Williamson, Hindy Najman, and Benjamin Ziemer who responded to an earlier draft of this article. We also thank the academic editor and the reviewers for their feedback. For the images of 1QIsa[a] from the Brill collection we are grateful to Brill Publishers. For the high-resolution, multi-spectral images of the Dead Sea Scrolls we are grateful to the Israel Antiquities Authority (IAA), courtesy of the Leon Levy Dead Sea Scrolls Digital Library; photographer: Shai Halevi. We are very grateful to the staff of the IAA Dead Sea Scrolls Unit for their help and support for our The Hands that Wrote the Bible project.

## Author Contributions

**Conceptualization:** Mladen Popović, Maruf A. Dhali, Lambert Schomaker.

**Data curation:** Mladen Popović, Maruf A. Dhali, Lambert Schomaker.

**Formal analysis:** Mladen Popović, Maruf A. Dhali, Lambert Schomaker.

**Funding acquisition:** Mladen Popović.

**Investigation:** Mladen Popović, Maruf A. Dhali, Lambert Schomaker.

**Methodology:** Mladen Popović, Maruf A. Dhali, Lambert Schomaker.

**Project administration:** Mladen Popović.

**Resources:** Mladen Popović, Lambert Schomaker.

**Software:** Maruf A. Dhali, Lambert Schomaker.

**Supervision:** Mladen Popović.

**Validation:** Mladen Popović, Maruf A. Dhali, Lambert Schomaker.

**Visualization:** Mladen Popović, Maruf A. Dhali, Lambert Schomaker.

**Writing – original draft:** Mladen Popović, Maruf A. Dhali, Lambert Schomaker.

**Writing – review & editing:** Mladen Popović, Maruf A. Dhali, Lambert Schomaker.

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
