## [Decision Letter · Decision Letter 0]

29 Dec 2020

PONE-D-20-32856

Artificial intelligence based writer identification generates new evidence for the unknown scribes of the Dead Sea Scrolls exemplified by the Great Isaiah Scroll (1QIsaa)

PLOS ONE

Dear Dr. Popović,

Thank you for submitting your manuscript to PLOS ONE. After careful consideration, we feel that it has merit but does not fully meet PLOS ONE’s publication criteria as it currently stands. Therefore, we invite you to submit a revised version of the manuscript that addresses the points raised during the review process.

We look forward to receiving your revised manuscript.

Kind regards,

Diego Raphael Amancio

Academic Editor

PLOS ONE

Journal Requirements:

2. Please ensure that you refer to Figures 23 and 24 in your text as, if accepted, production will need this reference to link the reader to the figure.

3. We note you have included a table to which you do not refer in the text of your manuscript. Please ensure that you refer to Table 2 in your text; if accepted, production will need this reference to link the reader to the Table.

4. We note that Figure 2  in your submission contain copyrighted images. All PLOS content is published under the Creative Commons Attribution License (CC BY 4.0), which means that the manuscript, images, and Supporting Information files will be freely available online, and any third party is permitted to access, download, copy, distribute, and use these materials in any way, even commercially, with proper attribution. For more information, see our copyright guidelines: http://journals.plos.org/plosone/s/licenses-and-copyright.

(1) You may seek permission from the original copyright holder of Figure 2 to publish the content specifically under the CC BY 4.0 license.

Reviewers' comments:

Reviewer's Responses to Questions

**Comments to the Author**

1. Is the manuscript technically sound, and do the data support the conclusions?

Reviewer #1: Yes

Reviewer #2: Yes

2. Has the statistical analysis been performed appropriately and rigorously? 

Reviewer #1: Yes

Reviewer #2: Yes

3. Have the authors made all data underlying the findings in their manuscript fully available?

Reviewer #1: Yes

Reviewer #2: Yes

4. Is the manuscript presented in an intelligible fashion and written in standard English?

Reviewer #1: Yes

Reviewer #2: Yes

5. Review Comments to the Author

Reviewer #1: The authors extracted character features from the Hebrew characters of one of the Dead Sea Scrolls and performed clustering analyses of the characters to determine whether a single scribe or two scribes wrote the manuscript. Features such as Hinge reflect motion changes in the hand of the writer and thus are idiosyncratic to individual writers. There is a fairly clear transition between two sections of the manuscript, indicating two writers. The authors have performed thorough statistical tests. The process is a machine-based and statistical one, rather than just the opinion of a human analyst.

Reviewer #2: Overall authors' approach to Palaeography is sound, and the application of

imaging/image processing, statistics, and the analysis derived from them seems

well founded. The data visualizations provided seem to me to represent an advance

for Palaeography. This is certainly how we should be vizualizing character shape

data across manuscripts going forward. The Hinge Feature plots (figure 4 and 5),

the Fraglet plot (figure 6), and the combined plot (figure 7) are quite good

and (should be) easily understood by the non-computationally minded

palaeographer. I find the heatmaps (figure 10) to be interesting, and probably

much more compelling when viewed interactively (i.e., not as part of a pdf, but

with customized software for manipulating the composites and their links back to

parts of the manuscript). Have the authors considered releasing a visualiztion

tool along with the data, showing the interactive features of these composites?

The ability to auto-classify both hand styles (those that are so common they're

basically "fonts") and the number of anonymous scribes who contributed to the

production of one manuscript is desirable. It is a "signal strength" problem -

which signal is stronger, the variations from scribes, or variations attributed

to other factors?

As of now, paleographers still make these classifications with the trained eye

based on what can only be considered a rudimentary qualitative/quantitative analysis.

Rarely is an opinion regarding a classification presented in a scientific fashion;

it's just "I assign this classification" based on "these manuscript images".

So the approach of the paper is significant.

Overall this work is significant and I believe this paper should be accepted for

publication. However, I do have some points of critique that I believe the authors

should consider.

1. The apparatus laid out in the paper has value even if the conclusion ("there

were two scribes; the break point is HERE") is less definitive. Papers like this

are valuable in order to gain acceptance for the methodology; there can still be

some debate over granular and specific conclusions.

2. That being said, the use of the Great Isaiah Scroll as the primary data for

the experiment is interesting, but also a bit wanting: Did one or two scribes create

the scroll? The authors wade in to a context where there is no real consensus, as

it is recognized that both positions are possible. However, the distinct break, the

caesura, in the scroll between columns 27 and 28 is a strong indicator (I think)

that the "two scribes" hypothesis is very likely. Two different manuscripts, one

containing columns 1-27 and another for 28-54, seems likely. If each section were

produced at different periods of time, then it is very possible that 2 scribes were used.

Furthermore, the experiment clearly shows that 2 scribes can be argued for and defended

scientifically with the approach the authors outline.

3. The paper states very clearly that deviation in character shapes occurs even when

one scribe is present. So it can still be argued that the deviations that lead to the

authors' conclusion come from something other than the two-author hypothesis. What

about the same author, separated in time, after an injury? Possibly far fetched, but

an argument that can be made nevertheless. It would be instructive for the authors

to run the same process on a text of similar length but known to be produced by one

scribe. Comparing the results of the two would then perhaps be a better evaluation

of their process.

4. In the same vein [developing "control" data/context for the methodology], it would

be interesting to introduce noise into the processing pipeline to see how robust

or fragile the critical conclusions really are. In Lines 171-175 the authors mention

some corrections/transformations (by-hand corrections like the application of affine

transformations to overcome distortion because of the photography of non-flat and/or

shape-distorted surfaces). What is the influence of such by-hand attempts at improving

the input data? By introducing a series of automated/known corrections, one can

measure through the entire system what the sensitivities would be in the output

conclusions. Showing that the final Fermi-Dirac conclusions and the heatmaps are

invariant/insensitive to such changes would lead to further confidence in the findings.

5. And this is why the methodology is so very critical for acceptance. It enables

such "stress testing" and teasing out of the places in the chain where there is noise.

Once an automated computational chain is set up, one can run experiments to test particular

conclusions and the robustness of them in the face of variations and noise. The

robustness (or lack thereof) would help to reveal arguments about the fragile nature

of the results w.r.t. the input.

6. One "knit-picky" point: the authors characterize their approach in the title and

abstract as being based on "artificial intelligence" techniques, but the set of tools

used (PCA, image processing, structural metrics of the letter forms, heat maps) are

not necessarily part of an overall AI-inspired approach. People might feel a little

misled because they might be looking for Convolutional Neural Networks or something

that looks more like "deep learning."

6. PLOS authors have the option to publish the peer review history of their article (what does this mean?). If published, this will include your full peer review and any attached files.

Reviewer #1: **Yes: **Gordon S. Novak Jr.

Reviewer #2: **Yes: **W. Brent Seales

---

## [Author Response · Author response to Decision Letter 0]

28 Jan 2021

Reviewers' comments:

Reviewer's Responses to Questions

The four reviewer's responses to questions where all answered YES by the reviewers.

Review comments to the author:

Reviewer #1: The authors extracted character features from the Hebrew characters of one of the Dead Sea Scrolls and performed clustering analyses of the characters to determine whether a single scribe or two scribes wrote the manuscript. Features such as Hinge reflect motion changes in the hand of the writer and thus are idiosyncratic to individual writers. There is a fairly clear transition between two sections of the manuscript, indicating two writers. The authors have performed thorough statistical tests. The process is a machine-based and statistical one, rather than just the opinion of a human analyst.

Response: We thank reviewer #1 for the very positive assessment of our approach to the writer identification process being a machine-based and statistical one, rather than just the opinion of a human analyst, indicating two writers.

Reviewer #2: Overall authors' approach to Palaeography is sound, and the application of

imaging/image processing, statistics, and the analysis derived from them seems

well founded. The data visualizations provided seem to me to represent an advance

for Palaeography. This is certainly how we should be vizualizing character shape

data across manuscripts going forward. The Hinge Feature plots (figure 4 and 5),

the Fraglet plot (figure 6), and the combined plot (figure 7) are quite good

and (should be) easily understood by the non-computationally minded

palaeographer. I find the heatmaps (figure 10) to be interesting, and probably

much more compelling when viewed interactively (i.e., not as part of a pdf, but

with customized software for manipulating the composites and their links back to

parts of the manuscript). Have the authors considered releasing a visualiztion

tool along with the data, showing the interactive features of these composites?

The ability to auto-classify both hand styles (those that are so common they're

basically "fonts") and the number of anonymous scribes who contributed to the

production of one manuscript is desirable. It is a "signal strength" problem -

which signal is stronger, the variations from scribes, or variations attributed

to other factors?

As of now, paleographers still make these classifications with the trained eye

based on what can only be considered a rudimentary qualitative/quantitative analysis.

Rarely is an opinion regarding a classification presented in a scientific fashion;

it's just "I assign this classification" based on "these manuscript images".

So the approach of the paper is significant.

Response: We thank reviewer #2 for the appreciation of the significance of our work and of us advancing palaeography by our data visualizations. Regarding the reviewer’s suggestion to release a data visualization tool, we have indeed thought about this. In the future we hope to make a web-based application available for experts and students. For the publication now, we have created a Zenodo DOI link where we have uploaded the binarized images, feature files, and a script for 3D visualization for the point clouds in PCA space. We refer to this at the end of the Methods and materials section.

Reviewer #2: Overall this work is significant and I believe this paper should be accepted for

publication. However, I do have some points of critique that I believe the authors

should consider.

The further points of feedback raised by reviewer #2 are in line with our own approach and thinking on these issues.

Reviewer #2: The apparatus laid out in the paper has value even if the conclusion ("there

were two scribes; the break point is HERE") is less definitive. Papers like this

are valuable in order to gain acceptance for the methodology; there can still be

some debate over granular and specific conclusions.

Response: We are grateful for the acknowledgement of the significance of our methodology to palaeography. We also appreciate that the comments of reviewer #2 have contributed to further strengthening our conclusions. After performing additional analysis, including random elastic morphing on the column images and an additional application of the logistic model to another shape feature than the Fraglets (i.e., now also to the Hinge feature), the presence of two statistically different signal sources in the column series appears to be clear.

Reviewer #2: That being said, the use of the Great Isaiah Scroll as the primary data for

the experiment is interesting, but also a bit wanting: Did one or two scribes create

the scroll? The authors wade in to a context where there is no real consensus, as

it is recognized that both positions are possible. However, the distinct break, the

caesura, in the scroll between columns 27 and 28 is a strong indicator (I think)

that the "two scribes" hypothesis is very likely. Two different manuscripts, one

containing columns 1-27 and another for 28-54, seems likely. If each section were

produced at different periods of time, then it is very possible that 2 scribes were used.

Furthermore, the experiment clearly shows that 2 scribes can be argued for and defended

scientifically with the approach the authors outline.

Response: We are pleased with the reviewer’s acknowledgement that in a sound and scientific manner our experiment shows that two scribes can be argued for. Together with his further points 3-5 we have performed additional tests that have further demonstrated the robustness of our method and conclusion. 

We have clarified in the introduction that in scrolls scholarship it is far from there being a recognition in the field that both positions are possible from a palaeographic point of view. On the contrary, hardly any palaeographic substantiation whatsoever has been put forward, except for Ulrich/Flint. In terms of palaeography, we have found a new means to move forward and present new evidence for two scribes, and we are pleased with reviewer #2’s appreciation of this.

Reviewer #2: The paper states very clearly that deviation in character shapes occurs even when

one scribe is present. So it can still be argued that the deviations that lead to the

authors' conclusion come from something other than the two-author hypothesis. What

about the same author, separated in time, after an injury? Possibly far fetched, but

an argument that can be made nevertheless. It would be instructive for the authors

to run the same process on a text of similar length but known to be produced by one

scribe. Comparing the results of the two would then perhaps be a better evaluation

of their process.

Response: We fully agree with the line of thinking of reviewer #2. This is exactly why we perform our analyses and tests, to find out what best explains the evidence for a clear break. 

While it is correct that there is variability in character shapes even when there is one scribe this does not mean that the overall data is best explained by within-writer variability. One would then expect the point cloud distribution to be more random instead of two separate clusters, without points going into each other’s area. 

Also, the change-of-pen-argument is weakened because Hinge and Fraglet analyses tap into different sources of information (Fraglets contain the larger, complicated character fragments, while the Hinge feature concerns local curvature) and both methods point to a clear break in the data and separation of two clusters. 

Moreover, we have performed an additional statistical F test that demonstrates a difference of variances of distance such that the variance is higher in the second part of the manuscript. This is indicative of more variable writing patterns in columns 28-54 over against 1-27. 

So, although one cannot rule out completely that the clear separation between the two halves of the manuscript and the difference in variance of writing patterns in each half is because of a change of writing implement (a different pen), writing fatigue or some injury that the writer suffered when moving on to the second half of the manuscript, the more straightforward explanation is that a change in writers occurred. 

We have further clarified this in the Results section, and in the Discussion and conclusions section. 

We would love to perform the additional test the reviewer asks for, but this cannot be performed on another text from the scrolls with a known author because the authors are not known; this is exactly the outstanding issue in the field we have advanced through our innovative approach. However, all our tests have inherently performed verifications for a known writer if someone reluctantly considers that only the first half is written by one scribe and then ignores the second half. Even after adding noise (new tests in the revision phase), the writers’ clusters still hold, and the separability between writers still prevails. This outcome also cross-validates the system’s robustness of writer identification. Nevertheless, we will perform our analysis on other texts from the scrolls in the near future, but those are separate studies. Of course, our writer identification tools have been validated by other corpora, to which we refer (see the ‘Firemaker’ reference on p. 20).

Reviewer #2: In the same vein [developing "control" data/context for the methodology], it would

be interesting to introduce noise into the processing pipeline to see how robust

or fragile the critical conclusions really are. In Lines 171-175 the authors mention

some corrections/transformations (by-hand corrections like the application of affine

transformations to overcome distortion because of the photography of non-flat and/or

shape-distorted surfaces). What is the influence of such by-hand attempts at improving

the input data? By introducing a series of automated/known corrections, one can

measure through the entire system what the sensitivities would be in the output

conclusions. Showing that the final Fermi-Dirac conclusions and the heatmaps are

invariant/insensitive to such changes would lead to further confidence in the findings.

Response: Again, we completely agree with reviewer #2. This is why we have performed additional stress tests adding noise in the form of random elastic morphing, which show exactly that what the reviewer requests: the results are invariant/insensitive to the noise introduced in the data, they stay the same, demonstrating the robustness of our initial findings. We describe and clarify this in the Methods and materials section, in the Results section, and in the Discussion and conclusions section. We have added a GitHub link to the random morphing software.

Reviewer #2: And this is why the methodology is so very critical for acceptance. It enables

such "stress testing" and teasing out of the places in the chain where there is noise.

Once an automated computational chain is set up, one can run experiments to test particular

conclusions and the robustness of them in the face of variations and noise. The

robustness (or lack thereof) would help to reveal arguments about the fragile nature

of the results w.r.t. the input.

Response: Our additional tests have demonstrated the robustness of our approach and our results, confirming our conclusions regarding the data. See our answer above to point 4. The bistable configuration of writing styles is confirmed by an additional fit of the logistic model on the randomly morphed column images, in the original feature space (Fraglets), but now also for the Hinge and the Adjoined Hinge+Fraglets feature vector. All analyses confirm the presence of a switch point. For Fraglets the position of the switch point was confirmed, for the Hinge feature, it is estimated to occur a bit more to the right, but in any case with a high reliability (high r, low sigma).

Reviewer #2: One "knit-picky" point: the authors characterize their approach in the title and

abstract as being based on "artificial intelligence" techniques, but the set of tools

used (PCA, image processing, structural metrics of the letter forms, heat maps) are

not necessarily part of an overall AI-inspired approach. People might feel a little

misled because they might be looking for Convolutional Neural Networks or something

that looks more like "deep learning."

Response: While it is certainly true that not all tools used are necessarily AI-inspired, we believe that it is warranted to use “artificial intelligence” in the title and abstract to characterize our innovative digital palaeography approach as an overall AI-inspired approach:

- The BiNet algorithm for binarization is a Convolutional Neural Network, in other words deep learning

- The Hinge approaches are from Pattern Recognition, generally considered as a subdiscipline in AI

- The codebook approaches (Kohonen maps) are from AI

- This is AI in the sense that we wanted to minimize human influence, letting the data speak

- In the classification stage we avoided deep learning to avoid discussions concerning explainability

We have clarified this accordingly in the Materials and methods section.

Editor comments:

1. Please ensure that your manuscript meets PLOS ONE's style requirements, including those for file naming. The PLOS ONE style templates can be found at ...

Response: We have done so accordingly, we have used the PLOS ONE style template (LaTeX).

2. Please ensure that you refer to Figures 23 and 24 in your text as, if accepted, production will need this reference to link the reader to the figure.

Response: We have explicated the reference on p. 17 and also on p. 15.

3. We note you have included a table to which you do not refer in the text of your manuscript. Please ensure that you refer to Table 2 in your text; if accepted, production will need this reference to link the reader to the Table.

Response: We have taken up a reference to the table on p. 5.

4. We note that Figure 2 in your submission contain copyrighted images. ... We require you to either (1) present written permission from the copyright holder to publish these figures specifically under the CC BY 4.0 license, or (2) remove the figures from your submission.

Response: We have received from Suzanne Mekking, Senior Acquisitions Editor Old Testament and Qumran, from Brill Publishers, the original copyright holder, the completed Content Permission Form for publication of the original image and attach it to this revised submission.

We have also included the text in the figure caption of the copyrighted figure on p. 6.

---

## [Decision Letter · Decision Letter 1]

25 Mar 2021

Artificial intelligence based writer identification generates new evidence for the unknown scribes of the Dead Sea Scrolls exemplified by the Great Isaiah Scroll (1QIsaa)

PONE-D-20-32856R1

Dear Dr. Popović,

We’re pleased to inform you that your manuscript has been judged scientifically suitable for publication and will be formally accepted for publication once it meets all outstanding technical requirements.

Kind regards,

Diego Raphael Amancio

Academic Editor

PLOS ONE

Additional Editor Comments (optional):

Reviewers' comments:

Reviewer's Responses to Questions

**Comments to the Author**

1. If the authors have adequately addressed your comments raised in a previous round of review and you feel that this manuscript is now acceptable for publication, you may indicate that here to bypass the “Comments to the Author” section, enter your conflict of interest statement in the “Confidential to Editor” section, and submit your "Accept" recommendation.

Reviewer #2: All comments have been addressed

2. Is the manuscript technically sound, and do the data support the conclusions?

Reviewer #2: Yes

3. Has the statistical analysis been performed appropriately and rigorously? 

Reviewer #2: Yes

4. Have the authors made all data underlying the findings in their manuscript fully available?

Reviewer #2: Yes

5. Is the manuscript presented in an intelligible fashion and written in standard English?

Reviewer #2: Yes

6. Review Comments to the Author

Reviewer #2: I am eager to see this paper appear in PLOS ONE.

I still don't like the title as it is (leading with "AI-based writer identification" diminishes your contribution, it seems to me, which is to use those tools in a clever and robust way to make your arguments). I'm not going to suggest what the title should be, but you should consider revising it.

And here is a little story: when I was part of a project at the Marciana Library in Venice to digitize the Venetus A, we used a robotic arm with a non-contact (structured light) probe to build 3D models of every page that we also photographed. We thought that was cool, and it enabled some new work with the data. The "Wired Magazine" article reporting on it was titled: "Robot Scans Ancient Manuscript in 3-D". https://www.wired.com/2007/06/robot-scans-ancient-manuscript-in-3-d/

The picture was of one of my students. HE was scanning the manuscript, USING a robot arm (not even a "robot", really).

In the same way YOU the authors have put together this method; AI is a tool. Bravo for using it and other methods. Don't let the headlines be that the AI robots figured it out instead of you.

7. PLOS authors have the option to publish the peer review history of their article (what does this mean?). If published, this will include your full peer review and any attached files.

Reviewer #2: **Yes: **W. Brent Seales

---

## [Editor Report · Acceptance letter]

31 Mar 2021

PONE-D-20-32856R1 

Artificial intelligence based writer identification generates new evidence for the unknown scribes of the Dead Sea Scrolls exemplified by the Great Isaiah Scroll (1QIsa^a^ )  

Dear Dr. Popović:

I'm pleased to inform you that your manuscript has been deemed suitable for publication in PLOS ONE. Congratulations! Your manuscript is now with our production department. 

Kind regards, 

on behalf of

Dr. Diego Raphael Amancio 

Academic Editor

PLOS ONE